# GOLDILOCS: GENERAL OBJECT-LEVEL DETECTION AND LABELING OF CHANGES IN SCENES

**Almog Friedlander**
Reichman University

**Ariel Shamir**
Reichman University

**Ohad Fried**
Reichman University

## ABSTRACT

We propose *GOLDILOCS*: a novel zero-shot, pose-agnostic method for object-level semantic change detection in the wild. While supervised Scene Change Detection (SCD) methods achieve impressive results on curated datasets, these models do not generalize and performance drops on out-of-domain data. Recent Zero-Shot SCD methods introduced a more robust approach with foundational models as backbone, yet they neglect the 3D aspect of the task and remain constrained to the image-pair setting. Conversely, 3D-centric SCD methods based on 3D Gaussian Splatting (3DGS) or NeRFs require multi-view inputs, but cannot operate on an image pair. Our key insight is that SCD can be reformulated as a 3D reconstruction problem over time, where geometric inconsistencies naturally indicate change. Although previous work considered viewpoint difference a challenge, we recognize the additional geometric information as an advantage. *GOLDILOCS* uses dense stereo reconstruction to estimate camera parameters and generate a pointmap of the commonalities between input images by filtering geometric inconsistencies. Rendering the canonical scene representation from multiple viewpoints yields reference images that exclude changed or occluded content. Rigid object changes are then detected through mask tracking, while nonrigid transformations are identified using SSIM heatmaps. We evaluate our method on a variety of datasets, covering both pairwise and multi-view cases in binary and multi-class settings, and demonstrate superior performance over prior work, including supervised methods.

## 1 INTRODUCTION

Scene Change Detection (SCD) is a fundamental task in computer vision with broad applications in anomaly detection (Kruse et al., 2024), environmental monitoring (Khan et al., 2017), infrastructure management (Han et al., 2021), and autonomous navigation (Krawciw et al., 2024; Takeda et al., 2023). Given images of the same scene captured at different times, i.e $T_0$ and $T_1$, SCD aims to detect meaningful changes in the scene while remaining robust to illumination, viewpoint, and occlusion. Accurate SCD enables powerful downstream capabilities, such as updating geospatial databases (Taneja et al., 2011b) and lawful surveillance (Huwer & Niemann, 2000; Zhang et al., 2020).

Despite its utility, SCD remains challenging. Early deep learning methods cast it as supervised segmentation, predicting binary change masks from image pairs (Alcantarilla et al., 2018; Varghese et al., 2018; Sakurada et al., 2020; Chen et al., 2021), but required large, domain-specific annotated datasets. To mitigate this, semi- (Park et al., 2022), self-supervised (Ramkumar et al., 2022; Furukawa et al., 2020b), and synthetic-data approaches (Park et al., 2021; Sachdeva & Zisserman, 2023a) were proposed, though generalization of these methods remains limited. More recently, foundation models have been introduced: SAMCD (Ding et al., 2024) integrates SAM (Kirillov et al., 2023), while ZSSCD (Cho et al., 2025) leverages SAM2 (Ravi et al., 2024) for segmentation and DEVA (Cheng et al., 2023) for mask tracking. While promising, these methods are confined to 2D and struggle to disentangle true changes from viewpoint differences.

In this work, we reformulate SCD as a 3D reasoning task. Our key insight is that meaningful changes appear as geometric inconsistencies in reconstructions from time-separated views. If an object is added, removed, or moved between $T_0$ and $T_1$, then 3D reconstruction algorithms, which assume scenes are static, will fail to reconcile the region, yielding inconsistent geometry.

We introduce *GOLDILOCS* (General Object-Level Detection and Labeling Of Changes in Scenes), a zero-shot framework for object-centric SCD. Our proposed framework identifies changed objects and labels the semantic type of change, inspired by experience of the three bears of the titular fairytale. Moreover, *GOLDILOCS* can detect changes even with a single image at $T_0$ and a single image at $T_1$, given sufficient 3D overlap. Our method leverages a dense stereo 3D reconstruction model (Leroy et al., 2024) to estimate camera intrinsics, extrinsics, and per-pixel 3D structure. It then performs depth filtering to remove temporally inconsistent geometry and yields a canonical static reconstruction of the scene. By comparing each input image to the clean rendering and propagating masks with SAM2 (Ravi et al., 2024), *GOLDILOCS* can identify and categorize scene changes as object-level additions, removals, movements, or non-rigid transformations. The latter are detected via SSIM maps (Wang et al., 2004) highlighting local structural distortions across time.

Unlike prior work, *GOLDILOCS* is training-free, calibration-free, and generalizes to unconstrained, real-world imagery. We demonstrate state-of-the-art binary and multi-class change detection under zero-shot conditions on both image pairs and sets.

## 2 RELATED WORK

This section reviews prior work, from early image differencing approaches to emerging zero-shot and 3D representation-based methods. The discussion highlights how existing approaches balance label dependence, generalization, and geometric reasoning, motivating our proposed solution.

**Pre-Neural Network Methods.**  Earlier approaches to SCD included simple image differencing, likelihood ratio tests, probabilistic mixture models, and extended up to shading models and background modeling; see Radke et al. (2005) for a survey. While these early methods established the systematic frameworks for change detection, their performance was often undermined by the reliance on pre-processing steps that addressed illumination variations and geometric misalignment.

**Supervised Methods**  Supervised SCD methods rely on labeled image pairs and change maps. Early work (e.g. Sakurada & Okatani (2015)) combined CNN features with superpixels. More advanced designs include ChangeNet (Varghese et al., 2018), a Siamese-inspired CNN (Mueller & Thyagarajan, 2016; Rao et al., 2017), and transformer-based models (Wang et al., 2021; Chen et al., 2021) exploiting attention for feature differencing. SimSaC (Park et al., 2022) adds an optical-flow warping module, while C-3PO (Wang et al., 2023) utilize a backbone network and trains separate branches for change types. CYWS-3D (Sachdeva & Zisserman, 2023b) extends supervision into 3D with a frozen transformer backbone. These methods perform well in-domain but remain tailored to dataset-specific styles and image pair relationships, limiting generalization.

**Semi-, Weakly- and Self-Supervised Methods**  To mitigate reliance on labeled data, semi-supervised (Lee & Kim, 2024), weakly-supervised (Sakurada et al., 2020) have been proposed. Additionally, SACD-Net (Furukawa et al., 2020a) learns in a self-supervised manner from unlabeled data by jointly optimizing viewpoint alignment and change detection. While these approaches reduce label cost, they often overlook the difficulty of collecting image pairs. Data augmentation (Sachdeva & Zisserman, 2023b; Lee & Kim, 2024) and synthetic datasets (Park et al., 2021) help increase training volume, but supervised pipelines still struggle with out-of-domain images, as robustness to style and domain variations is rarely addressed.

**Zero-Shot Methods.**  Zero-shot approaches aim to detect changes without relying on task-specific training data, but such methods remain scarce. ZSSCD (Cho et al., 2025) combines SAM2 (Ravi et al., 2024) for segmentation with DEVA (Cheng et al., 2023) for temporal tracking across image and video pairs. Nevertheless, these methods lack a 3D component to address viewpoint differences between $T_0$ and $T_1$.

**3D Representation-Based SCD**  Recent works approach SCD by explicitly modeling the 3D structure of a scene, enabling view-independent reasoning rather than direct image differencing. C-NeRF (Huang et al., 2023) detects changes by generating two NeRFs for $T_0$ and $T_1$, then comparing rendered views from aligned camera poses. Direction-consistent radiance differences highlight changes, but the method requires many images at both $T_0$ and $T_1$, controlled camera trajectories, and time-intensive optimization. 3DGS-CD (Lu et al., 2025) offers a more efficient alternative by

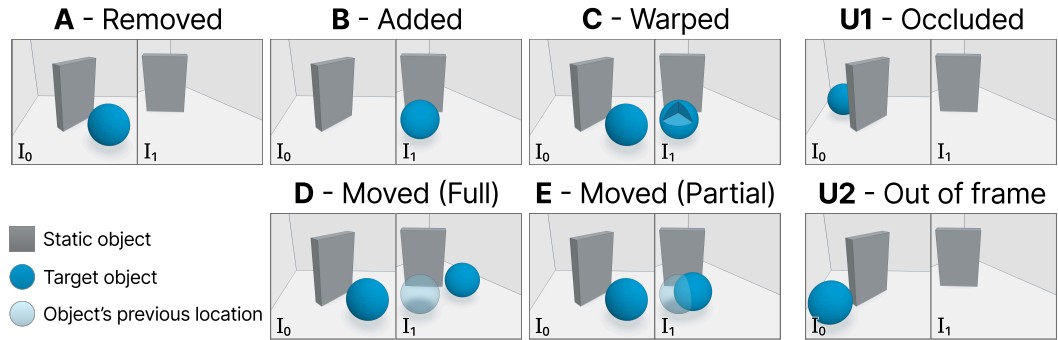

Figure 1: **Change types.** Scenes A and B illustrate object removal and addition, respectively. Scene C depicts a warped object that remains static in space but undergoes internal structural change. Scenes D and E illustrate full and partial object movements, distinguished by whether the object's volumes intersect between $T_0$ and $T_1$ (the light blue ghost ball is the old position at $T_0$ and is placed in the $I_1$ image for clarity). Scene U1 shows a case where the object's location is not visible in both views, and scene U2 depicts an occluded object—both are thus unclassifiable.

reconstructing a 3D Gaussian Splatting model from pre-change images and comparing novel-view renders against post-change observations with EfficientSAM (Xiong et al., 2023). This approach improves scalability and is robust to sparse post-change inputs, though it still requires a large image set taken at $T_0$, auxiliary images from $T_1$, and explicit reconstruction. Gaussian Difference (Jiang et al., 2025) extends this idea by encoding spatial and temporal information within a unified 4D Gaussian model. Instance IDs assigned via SAM and DEVA are used to track objects and identify temporal inconsistencies, enabling gaussians to be classified as changed or unchanged. This produces accurate change maps from arbitrary viewpoints while improving robustness to lighting variations and reducing computational cost, though multi-view image sets at both time points are still required.

Several related methods, such as Taneja et al. (2011a) and Palazzolo & Stachniss (2018), rely on a 3D model as the base , and detect changes in images by reprojecting views using the provided 3D geometry. Adam et al. (2022) extends this by requiring 3D models at both times, employing a graph optimization approach for object-level change detection. While these methods achieve strong performance, they depend on access to a 3D model at one or both times and cannot operate on RGB image inputs solely.

**Our Approach.** Building upon these advancements, our approach integrates the strengths of foundational models, 3D reasoning, and flexibility to varying inputs to achieve robust and efficient scene change detection. Using both 3D reconstruction and label-agnostic segmentation and tracking backbones, our method addresses the challenges of viewpoint variations and in-the-wild images, paving the way for more generalized and scalable SCD solutions.

## 3 PROBLEM DEFINITON

Given a pair of RGB images of a 3D scene, $I_0$ and $I_1$ captured at times $T_0$ and $T_1$ respectively, our goal is to detect and classify object-level changes between the two observations. Specifically, we aim to identify which objects have been *moved*, *added*, *removed*, or *warped* (i.e., undergone non-rigid transformations). For simplicity we demonstrate on the case of a pair of images, but these definitions naturally extend to a pair or sets of images, $\mathcal{I}_0$ and $\mathcal{I}_1$.

**Object-centric Change Detection Taxonomy** We propose a standardized taxonomy for object-level change labeling in 3D scenes, grounded in geometric and visibility-based reasoning. We define an object as any visually discernible entity present in an image. This includes volumetric elements such as boxes or furniture, as well as planar or texture-bound features like a blot of ink on a sheet of paper. The essential criterion is visibility and perceptual distinguishability in the image domain, irrespective of semantic identity or material composition. Some caveats to this definition include phenomena such as shadows, reflections, and camera-induced artifacts like lens flare, which, while not being objects in the conventional sense, are treated as such under this visibility-centric interpretation.

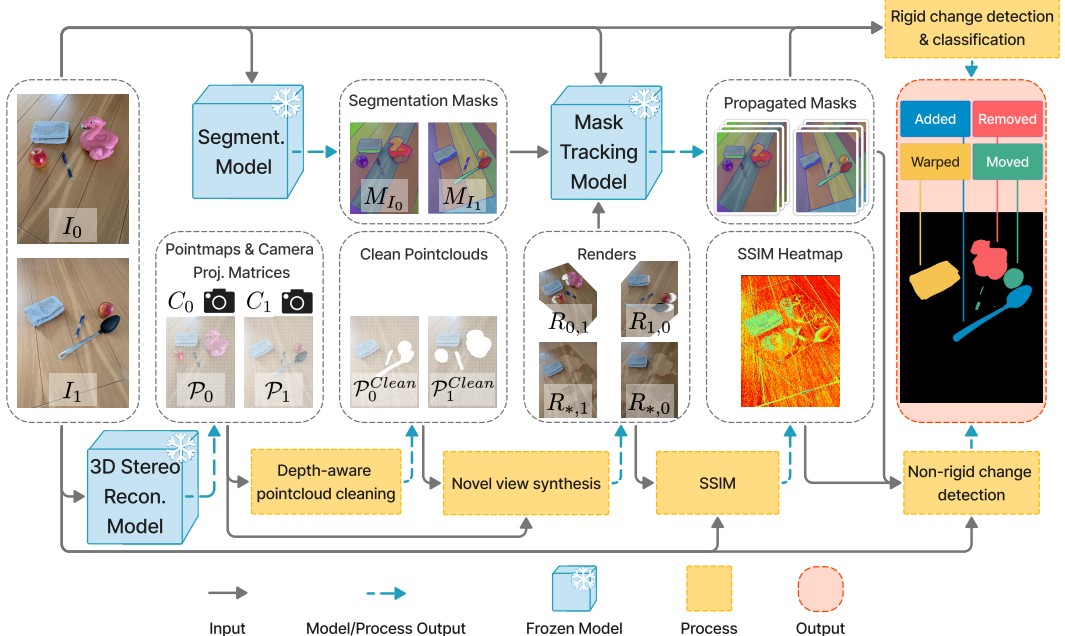

Figure 2: **Pipeline overview.** Given a pair of images, *GOLDILOCS* first reconstructs the 3D geometry of the scene, then uses rendering of novel views, segmentation, and tracking of object masks across time to identify object-level changes including non-rigid transformations.

Based on this definition, we categorize object-level changes into the following mutually exclusive types, and visualize them in Figure 1:

- **Removed.** An object that is visible in $I_0$ but no longer visible in $I_1$, and its spatial location at time $T_0$ is visible in $I_1$ (Figure 1A). If the region is occluded (Figure 1U1), or out of the frame (Figure 1U2) in $I_1$, we conservatively refrain from classifying the object as removed due to the absence of confirmatory evidence.

- **Added.** Symmetric to the removed case. An object labeled as removed in $I_0 \to I_1$ is labeled as added in $I_1 \to I_0$.

- **Moved.** An object that appears in both $I_0$ and $I_1$, but its location in 3D space has changed.[1] If the spatial volumes it occupies at $T_0$ and $T_1$ are disjoint, we term the move "full" (Figure 1D), and if they overlap, the move is "partial" (Figure 1E).

- **Warped.** An object is warped if it is visible in both $I_0$ and $I_1$, occupies approximately the same location in 3D space, but exhibits non-rigid deformation, e.g., bending, creasing, or squashing (Figure 1C).

## 4 METHOD

A key insight motivating our approach is that change detection between images can be naturally interpreted as a form of 3D reconstruction over time. However, while traditional multi-view stereo assumes that images are captured within a short time frame from different viewpoints, ensuring that the underlying scene remains static, we consider the case where the input images $I_0$ and $I_1$ are taken at different times, $T_0$ and $T_1$, such that the scene is allowed to undergo structural changes. To simplify the explanation, we first describe our framework assuming an image pair input; later, we will extend it to a pair of multi-view image sets, as detailed in Section 4.6.

Under this formulation, objects or regions that have changed between $T_0$ and $T_1$ will violate the static scene assumption. As a result, these regions are likely to produce inconsistent correspondences, contradicting depth estimates, or geometrically incompatible projections. By explicitly analyzing the

---

[1] We acknowledge potential ambiguity in cases involving visually identical instances (e.g., two identical pens, one removed, and another added to the scene), and assign a movement classification regardless.

3D reconstruction under temporal variation, we gain a principled and geometry-aware mechanism for localizing and identifying scene changes. An overview of our method can be found in Figure 2.

## 4.1 STEREO 3D RECONSTRUCTION AND CAMERA ESTIMATION

We employ MASt3R (Leroy et al., 2024), a foundational stereo 3D reconstruction model, as the core module to establish dense, geometrically grounded correspondences between a pair of images $I_0$ and $I_1$, to estimate the intrinsic and extrinsic parameters of the cameras, and to model the 3D structure of the scene. Specifically, given a pair of RGB images $I_0, I_1 \in \mathbb{R}^{H \times W \times 3}$, the stereo 3D reconstruction model produces dense pointmaps $\mathcal{P}_0, \mathcal{P}_1 \in \mathbb{R}^{H \times W \times 3}$ representing the 3D coordinates of each pixel, reconstructed in a shared coordinate system. The reconstruction also produces camera projection matrices $C_0, C_1 \in \mathbb{R}^{3 \times 4}$ with $C_i = K_i[R_i \,|\, t_i]$, where $K_i$ and $[R_i, t_i]$ are, respectively, the intrinsic and extrinsic camera parameters.

The choice of MASt3R for stereo reconstruction is motivated by its state-of-the-art performance on uncalibrated images and its ability to produce dense geometry and camera poses, key to our 3D reasoning. However, this component can be replaced by any other method with similar outputs.

## 4.2 CONFLICT RESOLUTION VIA DEPTH FILTERING

To eliminate geometric inconsistencies and occluders, we clean the pointmaps using a reverse depth test. Let $\mathcal{P}_i$ denote the pointmap reconstructed from image $I_i$, where each entry is a 3D point in world coordinates. For every target view $I_j$, we additionally have a depth map $D_j$ that records the camera-space $z$ value at each pixel, derived directly from the reconstruction process.

Given a world point $p \in \mathcal{P}_i$, we reproject it into the image plane of image $I_j$ to obtain the pixel coordinates $(u, v)$ and its camera-space depth $z_{i \to j}$:

$$(u, v, z_{i \to j}) = C_j p$$

where $C_j$ is the projection operator (intrinsics and extrinsics) mapping world-space points to image coordinates in view $j$.

**Depth Conflict Resolution.**  We define the cleaned pointmap $\mathcal{P}_i^{Clean} \subseteq \mathcal{P}_i$ as the subset of points from $\mathcal{P}_i$ that do not conflict with the geometry observed in the opposing view $I_j$. Specifically, a point $p \in \mathcal{P}_i$ is considered *conflicting* with view $j$ if:

$$z_{i \to j} < D_j(u, v), \quad \text{where } (u, v, z_{i \to j}) = C_j p$$

This condition indicates that $p$ would project in front of the surface observed by $I_j$, and therefore represent an occluder or a changed object.

By removing geometrically conflicting points through reverse depth filtering, we obtain $\mathcal{P}_0^{Clean}$ and $\mathcal{P}_1^{Clean}$ that exclude points that are likely to correspond to objects that were added, removed, or moved between $T_0$ and $T_1$. The union of these cleaned pointmaps, denoted $\mathcal{P}_*$, is composed of the farthest visible surfaces of the scene and provides a consistent representation of the static scene geometry. A detailed example of this process is illustrated in Figure 8 in the appendix.

**Rendering Pointmaps as Images.**  We denote by $R_{i,j}$ the rendered image of a pointmap $\mathcal{P}_i$ from the viewpoint of $I_j$, where each point $p \in \mathcal{P}_i$ is projected using $C_j$ and colored using its appearance in image $I_i$. For example, $R_{0,0}$ denotes a rendering with the geometry and viewpoint of $I_0$, an approximate reconstruction of the original image $I_0$, while $R_{0,1}$ synthesizes how the geometry captured in $I_0$ would appear from the viewpoint of $I_1$, allowing cross-time and cross-view comparisons.

The joint cleaned pointmap $\mathcal{P}_*$ is utilized as a canonical reference geometry for downstream comparisons. For example, to identify objects only visible in $I_0$, we render $\mathcal{P}_*$ using $C_0$, producing $R_{*,0}$. Discrepancies between this rendering and the reference image $I_0$ indicate the presence of dynamic or changed elements. By analyzing such discrepancies in the image space, we can reliably localize and classify the types of changes the objects underwent.

### 4.3 SEMANTIC SEGMENTATION

We apply semantic segmentation through a foundational model for class-agnostic dense segmentation (Ravi et al., 2024), which offers state-of-the-art instance segmentation and propagation. As with the 3D reconstruction model, this component is modular and any mask-generating and tracking system can be used instead.

For an image $I_i$, the segmentation model outputs a set of binary masks:

$$M_{I_i} = \{mask_1, mask_2, \ldots, mask_k\}, \quad mask \in \{0,1\}^{H \times W} \tag{1}$$

To produce object-level predictions that align with our change taxonomy, we operate on rendered views of the reconstructed scene. Specifically, to detect objects that were **removed** or **moved** between times $T_0$ and $T_1$, we first segment the rendering $R_{0,1}$, obtaining masks $M_{R_{0,1}}$. Note that we segment $R_{0,1}$ and not $I_0$ as most often ground truth is annotated from the point of view of $I_1$, thus it is required to generate masks for objects present only in $I_0$, e.g removed objects, using $C_1$ to achieve the desired predictions. Masks are then propagated to the clean rendering $R_{*,1}$ using the object tracking functionality of the segmentation model. Masks that fail to track are considered changed:

$$M_{R_{0,1}}^{\text{Changed}} = M_{R_{0,1}} \setminus \text{Track}(M_{R_{0,1}}, R_{0,1} \to R_{*,1})$$

We then pass the masks $M_{R_{0,1}}^{Changed}$ to the tracking model and propagate from $R_{0,1}$ to $I_1$ to differentiate between moved and removed objects:

$$M_{R_{0,1}}^{Moved} = \text{Track}(M_{R_{0,1}}^{Changed}, R_{0,1} \to I_1)$$

$$M_{R_{0,1}}^{Removed} = M_{R_{0,1}}^{Changed} \setminus M_{R_{0,1}}^{Moved}$$

To identify **added** objects and validate **moved** objects detected in the previous process we perform a complementary procedure:

$$M_{I_1}^{\text{Changed}} = M_{I_1} \setminus \text{Track}(M_{I_1}, I_1 \to R_{*,1})$$

$$M_{I_1}^{Moved} = \text{Track}(M_{I_1}^{\text{Changed}}, I_1 \to R_{0,1})$$

$$M_{I_1}^{Added} = M_{I_1}^{\text{Changed}} \setminus M_{I_1}^{\text{Moved}}$$

**Filtering.** To prevent false positives in occluded or out-of-view regions, we apply a visibility-aware filtering step to all tracked mask sets. For each predicted mask, we compute the proportion of its area that overlaps with gaps in $R_{0,1}$, i.e., regions that are either occluded or lie outside the mutually visible field of view between the two input images. We discard masks with an overlap proportion larger than a threshold $\alpha \in [0, 1]$. This filtering step ensures that only masks grounded in reliable geometric evidence are retained. The value of $\alpha$ is set once for each dataset, and correlates to the amount of overlap between image pairs, see appendix A.6 for detailed explanation.

### 4.4 DETECTING NON-RIGID CHANGES VIA VIEWPOINT-ALIGNED COMPARISON

Beyond rigid changes; additions, removals, and movement, we identify static objects that experienced non-rigid transformations that alter their structure. Towards this end we compute a dense *Structural Similarity Index Measure* (SSIM) map between the rendering $R_{0,1}$ and the target image $I_1$. SSIM is well suited for this task as it captures perceptual differences in structure, luminance, and contrast, making it particularly effective in detecting subtle appearance changes in deformable objects that remain spatially static, such as a tablecloth that appears smooth in $T_0$ but wrinkled in $T_1$. We calculate the mean dissimilarity score of each static segmentation mask in $M_{I_1}$, and classify those farther than a standard deviation from the mean across all masks as **warped**.

### 4.5 PER PIXEL PREDICTION

The per-pixel prediction for the target view $I_1$ is obtained by stacking the filtered labeled masks according to the priority order $Warped > Moved > Removed > Added$. This ordering ensures that removed-object masks take precedence over added-object masks, resolving cases where an object may appear "added" simply because it was previously occluded by a removed object rather than newly introduced into the scene.

### 4.6 MULTI-VIEW GENERALIZATION

Although introduced for a pair of images, our method naturally extends to a pair of image *sets* consisting of multiple views at either or both times: $\mathcal{I}_0 = \{I_0^j\}_{j=1}^n$ from $T_0$ and $\mathcal{I}_1 = \{I_1^i\}_{i=1}^m$ from $T_1$. This capability expands *GOLDILOCS*'s scope from image pairs to multi-view scenarios, including cases with a single image from $T_1$, where $|\mathcal{I}_1| = 1$.

**Top-1 Selection via Geometric Matching.**   We define the matching score scalar value of two images as the number of corresponding points that are matched between the images by the 3D stereo reconstruction model. We compute pairwise matching scores between every target view $I_1^i \in \mathcal{I}_1$ and each candidate view $I_0^j \in \mathcal{I}_0$. Then, each $I_1^i$ is paired with its top ranked match from $\mathcal{I}_0$ as input for the pairwise pipeline as before.

**Cross-View Voting via Pixel Correspondence.**   Let $\hat{I}^i$ be the per-pixel predicted labels for $I_1^i$, where each pixel is labeled from the set $\mathcal{L} = \{Added, Removed, Moved, Warped, Unchanged\}$. Given a segment $s$ in $\hat{I}^i$ whose label is $l \in \mathcal{L}$, let $\mathcal{P}_i(s) \in \mathcal{P}_i$ be the set of 3D points that projects to $s$. We map the coordinates of the pixels of $s$ into $\hat{I}^j$ by projecting $\mathcal{P}_i(s)$ to the set of visible pixels in $C_j$, denoting the labels of this projected region by $\hat{I}^j(s_{i \to j})$. Each different view $I_1^j \in \mathcal{I}_1$ then votes on the label of $s$ according to the per-pixel majority vote within $\hat{I}^j(s_{i \to j})$:

$$label_i(s) = l, \qquad label_{j(j \neq i)}(s) = \mathrm{mode}(\hat{I}^j(s_{i \to j}))$$

If $|\mathcal{I}_1| = m$, then the final label of segment $s$ is given by cross-view majority:

$$\widehat{label}(s) = \mathrm{mode}\left(\{label_j(s) \mid j = 1, \ldots, m\}\right).$$

Ties are resolved by a fixed priority over $\mathcal{L}$. This ensures that each segment casts exactly one vote per view towards the final prediction, leading to consistent labeling across all target images. More details can be found in Appendix A.5, and a visualization of the additional components relating to the multi-view extension and it's interface with the pairwise pipeline in Figure 11.

## 5 RESULTS

We evaluate our method across a diverse set of datasets including both real and simulated environments, indoor and outdoor scenes, and various types of change. These datasets differ in image alignment, annotation granularity, and in the types of changes they contain: *target changes*, which correspond to the object-level modifications we aim to detect, and *non-target changes*, which arise from imaging conditions such as illumination, weather, or seasonal variation. While *GOLDILOCS* relies on a segmentation model that requires parameters, most settings are shared across datasets; we did not conduct a brute-force parameter search for the optimal settings for each dataset, and instead recommend values derived from dataset characteristics. See Appendix A.6 for further details.

A summary of these key characteristics is provided in Table 1, and an in depth explanation is provided in Appendix A.1. Following the standard evaluation protocols commonly used for each dataset, we report Precision, Recall, F1-score, and Intersection over Union (IoU) for the NeRFCD and 3DGS-CD datasets. For ChangeSim and VL-CMU-CD, we additionally compute per-class IoU and F1, along with mean IoU and macro-averaged F1.

We evaluate our framework against a broad set of baseline methods, reporting performance using each method's original evaluation protocol. On datasets that include both binary and multi-class labels, we assess our approach under both settings to provide a comprehensive evaluation. A method is considered *in-domain* if it was trained on $T_0$–$T_1$ image pairs from the dataset prior to inference on the test set. Table 2 summarizes the capabilities of all compared methods, including required training, support for image pair and multi-image inputs, semantic change labeling, and object-level prediction. Qualitative results of our method are presented in Figure 3, and examples of the warped change type appear in Figure 6 in the appendix.

Table 1: Overview of the evaluation datasets.

| Dataset | Setting | Real / Sim | Indoor / Outdoor | Image Alignment | Target Change | Non-target Change |
|---|---|---|---|---|---|---|
| VL-CMU-CD (Alcantarilla et al., 2018) | Pairwise | Real | Outdoor | Coarse | Missing objects | Weather, illumination |
| RC-3D (Sachdeva & Zisserman, 2023b) | Pairwise | Real | Indoor | None | Missing objects | Illumination |
| ChangeSim (Park et al., 2021) | Pairwise | Simulated | Indoor | None | New, missing, moved, replaced | Illumination |
| NeRFCD (Huang et al., 2023) | Multi-view | Real | Indoor | None | New objects | Illumination |
| 3DGS-CD (Lu et al., 2025) | Multi-view | Real | Indoor & Outdoor | None | New, moved objects | Illumination |

Table 2: Comparison of baseline SCD methods.

| Method | Zero-shot | Pair-wise | Multi-image | Min. $T_1$ Views required for multi-image inference | Change Labeling | Warp Detection | Object-level Prediction |
|---|---|---|---|---|---|---|---|
| ChangeNet (Varghese et al., 2018) | | ✓ | | | | | ✓ |
| CSCDNet (Sakurada et al., 2020) | | ✓ | | | | | ✓ |
| CYWS-3D (Sachdeva & Zisserman, 2023b) | | ✓ | | | | | ✓ |
| DR-TANet (Chen et al., 2021) | | ✓ | | | ✓ | | ✓ |
| C-3PO (Wang et al., 2023) | | ✓ | | | ✓ | | ✓ |
| ZSSCD (Cho et al., 2025) | ✓ | ✓ | | | ✓ | | ✓ |
| C-NERF (Huang et al., 2023) | ✓ | | ✓ | 20+ | | | |
| 3DGS-CD (Lu et al., 2025) | ✓ | | ✓ | 4 | ✓ | | ✓ |
| Gaussian Diff. (Jiang et al., 2025) | ✓ | | ✓ | 20+ | | | ✓ |
| **GOLDiLOCS** (ours) | ✓ | ✓ | ✓ | **1** | ✓ | ✓ | ✓ |

**ChangeSim**    Table 3 presents quantitative results on the ChangeSim dataset, evaluating performance in both binary and multi-class settings. *GOLDILOCS* outperforms all baselines in terms of IoU and mIoU for binary change detection, and achieves superior results across all multi-class categories and mean IoU—except for the *rotated* class. Although we map ChangeSim's *rotated* category to our *moved* change type, in practice, many objects labeled as *rotated* are only slightly reoriented or toppled without significant spatial displacement. Given the dataset's resolution, scene depth, and the small pixel footprint of individual objects, this class remains particularly challenging for our framework. Table 8 further reports F1 scores on the binary changed class alone (a less commonly used protocol) where *GOLDILOCS* still outperforms several supervised, in-domain baselines.

**VL-CMU-CD**    While *GOLDILOCS* does not match the performance of supervised, in-domain baselines, we attribute this to two key factors: (1) our framework produces precise object-level masks, whereas the dataset's ground truth consists of coarse polygonal annotations, and (2) the dataset contains noisy labels with some genuine changes left unannotated, further penalizing methods that make fine-grained predictions, e.g., Figure 3 row 1 on the right. Despite these challenges, *GOLDILOCS* achieves F1 score of 61.8%, the highest score among zero-shot methods, outperforming ZSSCD by a significant margin. Detailed results can be found in Table 8 in the appendix.

**RC-3D**    *GOLDILOCS* outperforms all baselines on this dataset while relying solely on RGB input and requiring no training, achieving an mAP of 0.531. In comparison, CYWS using only RGB input reaches mAP of 0.14, while CYWS-3D significantly boosts performance to 0.41, and further improves to 0.50 with RGB-D input. We attribute our method's performance to the strong geometric priors encoded by the 3D reconstruction model, which enables our method to localize object-level changes downstream with high precision, even in the absence of depth input or dataset-specific training. Detailed results are provided in Table 9 in the appendix.

**3DGS-CD**    Table 4 reports performance on the 3DGS-CD dataset across five diverse scenes. Note that *GOLDILOCS* performs inference on the test set without access to additional $T_1$ views, while 3DGS-CD baseline relies on a separate subset of four $T_1$ images for reconstruction and inference. Moreover, unlike 3DGS-CD, our method avoids time-intensive explicit 3D reconstruction, achieving test-time inference within 5 minutes per image compared to the hours required by the full 3DGS-CD pipeline. Even without auxiliary $T_1$ frames and despite its significantly faster runtime, *GOLDILOCS* outperforms in both IoU and F1 metrics.

**NeRFCD**    Table 4 presents quantitative results on scenes from the NeRFCD dataset. The compared methods achieve strong performance, but rely on dozens to hundreds of images per time point, captured from overlapping camera trajectories, and require significant computational effort. In contrast, *GOLDILOCS* outperforms other methods in the F1 and IoU metrics while operating in a zero-shot setting, without auxiliary $I_1$ images, in substantially shorter runtimes as shown in Table 5.

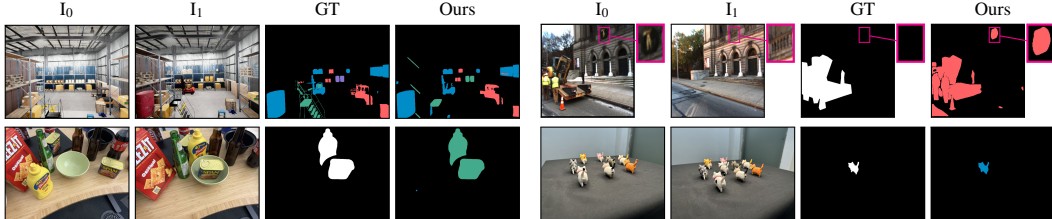

Figure 3: Examples of different change types (Added, Removed, Moved, Replaced) detected by our method compared to ground truth (GT) in indoor and outdoor scenes. Our results provide color-coded labels even with binary GT and can exceed its accuracy, as annotations are often coarse or miss small changes. This, however, can lower our reported metrics as we compare to such imperfect GT. Figure 5 provides more qualitative results.

Table 3: Results (IoU) on the ChangeSim dataset comparing binary and multi-class performance. mIoU provides the holistic metric, averaging over all multi-class labels.

| | Binary | | | Multi-class | | | | | |
|---|---|---|---|---|---|---|---|---|---|
| Method | Changed | Unchanged | mIoU | Added | Removed | Moved | Replaced | Unchanged | mIoU |
| ChangeNet | 17.6 | 73.3 | 45.4 | 9.1 | 6.9 | 11.6 | 6.6 | 80.6 | 23.0 |
| CSCDNet | 22.9 | 87.3 | 55.1 | 12.4 | 6.0 | **17.5** | 7.9 | 90.2 | 26.8 |
| C-3PO | 28.8 | 90.4 | 59.6 | 13.3 | 8.0 | 16.9 | 8.0 | **92.6** | 27.8 |
| ZSSCD[†] | 25.2 | 89.3 | 57.2 | 21.9 | 9.6 | 0.0 | 12.6 | 89.3 | 26.7 |
| **Ours** | **37.3** | **92.5** | **64.9** | **25.4** | **20.9** | 7.7 | **21.4** | 92.5 | **33.6** |

[†] In order to perform an accurate comparison with ZSSCD which reports metrics on video sequence inputs, we evaluated their method on the dataset in a pair-wise setting utilizing their published code.

Table 4: Results on the NeRFCD and 3DGS-CD datasets. F1 combines precision and recall, providing a measure of overall performance, and can be regarded as the primary metric along with IoU.

| | | # $T_1$ Images | | | | | |
|---|---|---|---|---|---|---|---|
| Dataset | Method | Evaluation | Auxiliary↓ | Precision↑ | Recall↑ | F1↑ | IoU↑ |
| 3DGS-CD | 3DGS-CD | 4 | 4 | **97.84** | 97.60 | 97.51 | 95.16 |
| | **Ours** | 4 | **0** | 97.09 | **98.17** | **97.72** | **95.30** |
| NeRFCD | Gaussian Difference[†] | 10 | 10+ | **95.91** | 88.83 | 91.90 | 85.74 |
| | D-NeRF | 10 | 10+ | 47.22 | 66.92 | 52.92 | 37.73 |
| | C-NERF | 10 | 10+ | 84.93 | **93.39** | 88.89 | 80.83 |
| | **Ours** | 10 | **0** | 95.14 | 89.65 | **91.97** | **87.62** |

[†] Note that Gaussian Difference averaged over only 4 of the scenes (Cats, Go pieces, Desk, Potting), omitting the Block scene.

Table 5: Runtime comparison (in minutes) across pipeline stages on the multi-image Potting scene from the NeRFCD dataset. Our method is over 10× faster than Gaussian Difference and nearly 100× faster than C-NeRF.

| Method | 3D Reconstruction | Change Detection | Rendering/Gaussian Partition | Total |
|---|---|---|---|---|
| C-NeRF | 964.2 min | 45.0 min | 46.8 min | 1056.0 min |
| Gaussian Difference | 70.2 min | 6.0 min | 72.0 min | 148.2 min |
| **Ours** | **4.7 min** | **5.2 min** | **0.6 min** | **10.6 min** |

## 6 ABLATIONS

We perform ablation experiments to highlight the importance of our 3D reasoning, segmentation and cross-view voting components. In ChangeSim (Table 10), removing stereo reconstruction and novel view synthesis significantly reduces IoU, particularly in the removed category, confirming that accurate 3D geometry contributes to reliable change localization. An additional analysis on 3DGS-CD (Figure 9) shows how performance degrades as $I_0$ images with fewer geometric correspondences are

used as matches. Both F1 and IoU trend downwards with image rank, and reconstruction eventually fails below the 20th percentile. Moreover, ablation of the cross-view voting component (Table 11) shows that it yields large gains in precision and IoU by filtering noisy detections and enforcing consistency across views. We conducted an experiment to examine the relationship between 3D reconstruction quality and detection performance. As shown in Figure 10, even with only 5% of the full reconstruction steps, nearly 90% of the final mIoU is retained, demonstrating robustness to varying reconstruction quality. Finally, see Table 13 for an analysis of how different detection thresholds of the segmentation component affect performance on the VL-CMU-CD dataset.

## 7 LIMITATIONS

*GOLDILOCS* relies on SAM2 (Ravi et al., 2024) as a foundational model, which can misinterpret shadows, reflections, texture patterns, or camera artifacts (e.g., lens flares, specular highlights and motion blur of fast moving elements) as objects due to strong local contrast or 3D projection. Segmentation models are also parametric, requiring different settings for optimal performance across scenarios. Object scale, for example, may vary drastically—a tennis ball might fill half the image or occupy only a few pixels depending on scene depth—necessitating different sensitivity values. Lighting variations also affect other components, as drastic illumination shifts can introduce SSIM errors or hinder 3D reconstruction. Additionally, the reliance on statistical SSIM thresholds might lead to misclassifications under certain conditions and more sophisticated techniques may further improve detection on edge cases.

Finally, the prediction pipeline employs a label priority (Section 4.5) to resolve overlapping categories. As a result, nested changes within larger objects are not independently labeled: for instance, if a keyboard is displaced between $T_0$ and $T_1$ but a key is also removed, the mask relating to the keyboard is labeled as *moved*, while the removed key is suppressed.

## 8 CONCLUSION

To solve the problem of object-level scene change detection, we introduced *GOLDILOCS* — a zero-shot, pose-agnostic framework for both pairwise and multi-view inputs that reformulates SCD as a 3D reconstruction problem over time. Our method decouples the challenges of 3D reconstruction from the semantic interpretation of change, allowing each component to be addressed using foundational models without further training. We fuse geometric and semantic cues in a modular inference pipeline, identify rigid object changes through mask propagation, and detect non-rigid transformations via SSIM-based comparison. This approach bypasses the limitations of 2D-only methods and generalizes beyond curated datasets and training regimes.

*GOLDILOCS* achieves strong zero-shot performance in one-to-one, multi-to-one and multi-to-multi settings across various real-world and synthetic datasets, while its generality and training-free approach allows change detection under minimal assumptions. Looking forward, integrating more expressive segmentation models or specialized 3D reconstruction techniques may further enhance performance. Future avenues of research include support for multi-label classes, nested objects and better distinction of non-target changes, enabling even more powerful general SCD.

## 9 REPRODUCIBILITY STATEMENT

The core implementation details, including the foundational models serving as the backbone for *GOLDILOCS*, are presented in Section 4, which also covers depth-aware point cloud cleaning, novel view synthesis and rendering, SSIM heatmap generation, and both rigid and non-rigid change detection. Prediction pipelines are described separately: the pairwise case in Section 4.5, the multi-view case in Section 4.6, and an extended version of the latter in Appendix A.5. Dataset usage appears in Section 5, with an extended version including pre-processing and related details provided in Appendix A.1. Finally, parameter settings and the rationale behind their selection are discussed in Appendix A.6. Additionally, we will release our code to support reproducibility and further research.

## 10 ACKNOWLEDGMENTS

This work was partially supported by the Israel Science Foundation Grant no. 1427/25, 1574/21 and Joint NSFC-ISF Research Grant no. 3077/23.

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

# A APPENDIX

## A.1 DATASET OVERVIEW

**ChangeSim** (Park et al., 2021) is a benchmark dataset for scene change detection, comprising unaligned image pairs from a simulated industrial indoor environment. It includes annotations of changes at the object level classified into four types: *new*, *missing*, *rotated*, and *replaced*. Although prior work has focused mainly on binary change detection, we use fine-grained labels in the data set to perform multiclass prediction. To align with our change taxonomy, which does not explicitly define a 'replaced' class, we interpret predicted object additions that spatially overlap with predicted object removals as instances of replacement. Similarly, the 'rotated' category is treated as equivalent to our definition of partially-moved objects. All experiments were carried out on the full set of 8,212 pairs of images at a native resolution of 640×480.

**VL-CMU-CD** (Alcantarilla et al., 2018) is a dataset for the detection of scene changes in the real world consisting of aligned and time-separated images. The dataset contains significant visual variations due to seasonal and illumination differences. Following prior work, we evaluate on the test set comprising of 429 image pairs, resized to a resolution of 512×512, and treat the task as binary change detection. Given that the dataset's ground truth contains annotations of objects that were present in $T_0$ and are missing in $T_1$, only the *removed* class in our change taxonomy is applicable, and we restrict evaluation to this category.

**RC-3D** (Sachdeva & Zisserman, 2023b) is a recently introduced dataset for change detection in real world indoor scenes consisting of unaligned image pairs. It includes bounding box annotations of changes at the object level. Following prior work, we evaluate on the available 100 image pairs provided at a resolution of 1920×1440, using the mean average precision metric defined by Everingham et al. (2010). Given that the dataset's ground truth contains annotations of objects that were present in $T_0$ and are missing in $T_1$, only the *removed* class in our change taxonomy is applicable, and we restrict evaluation to this category.

**3DGS-CD** (Lu et al., 2025) is a real-world scene change detection dataset captured in cluttered indoor environments. It consists of five scenes – *Mustard*, *Desk*, *Swap*, *Bench*, and *Sill* – each exhibiting object-level changes limited to additions and movements, with binary ground truth. For each scene, approximately 150–200 images are captured at time $T_0$ and 8 unaligned images are taken at time $T_1$, of which 4 are used for testing, provided at a resolution of 1008×756. An exception is *Sill*, which includes 60 $T_1$ unaligned images, with 30 reserved for testing.

**NeRFCD** (Huang et al., 2023) is a scene change detection dataset designed to evaluate object-level changes, including forward-facing and 360-degree surrounding setups. It contains ten scenes in total: two *surrounding* scenes (*Cats* and *Block*) and eight *forward-facing* scenes (*Go Pieces*, *Mural*, *Card*, *Text*, *Potting*, *Desk*, *Pottery* and *Sculpture*). For each scene, approximately 20–200 unaligned images are captured at times $T_0$ and $T_1$, of which 10 are reserved for testing. Although the images are unaligned, the cameras at both times follow similar motion trajectories to reduce viewpoint discrepancies. We conduct our experiments at a resolution of 1008×756. Due to the nature of the scenes and the subtlety of the changes involved, we excluded *Mural*, *Card*, *Text*, *Sculpture*, and *Pottery* from our experiments. As presented in Figure 4, these scenes focus on close-up views of individual objects, where changes are limited to fine surface-level details such as abrasions or ink specks, which are out-of-scope of our object-centric detection framework, and also do not fall under the *warped* change type from our taxonomy.

**3RScan** (Wald et al., 2019) is a 3D-centric real-world dataset of indoor environments designed to evaluate 3D object instance re-localization. It consists of 1482 RGB-D scans across 478 scenes. For each scan, the dataset provides calibrated RGB-D image sequences with corresponding camera intrinsics and extrinsics, globally aligned textured meshes, dense instance-level semantic annotations, as well as ground-truth object alignments describing rigid transformations and symmetries between matched instances. This evaluation of geometric and semantic changes in 3D such as object additions, removals, and movements. We conduct our experiments at a resolution of 960×540 and present qualitative results in Figure 7.

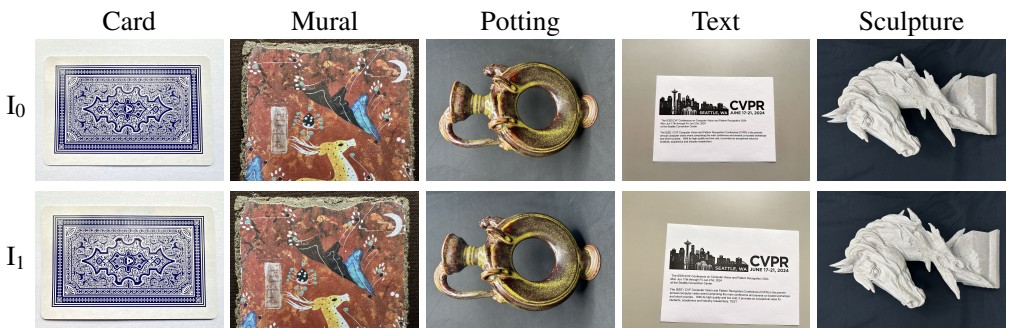

| | Card | Mural | Potting | Text | Sculpture |

Figure 4: Scenes excluded from the NeRFCD dataset.

Table 6: Detailed results on selected scenes from the NeRFCD dataset.

| Dataset | Method | Evaluation | # T₁ Images Auxiliary↓ | Precision↑ | Recall↑ | F1↑ | IoU↑ |
|---|---|---|---|---|---|---|---|
| Cats | Gaussian Difference | 10 | 115 | **97.47** | 93.21 | 92.41 | 91.03 |
| | D-NeRF | 10 | 115 | 59.24 | 60.57 | 59.68 | 42.71 |
| | C-NERF | 10 | 115 | 86.70 | 90.46 | 88.54 | 79.47 |
| | **Ours** | 10 | **0** | 95.79 | **97.91** | **96.76** | **93.83** |
| Block | D-NeRF | 10 | 177 | 25.83 | 79.48 | 38.06 | 23.88 |
| | C-NERF | 10 | 177 | 77.70 | **93.62** | 84.92 | 74.04 |
| | **Ours** | 10 | **0** | **89.11** | 82.38 | **85.42** | **81.59** |
| Go pieces | Gaussian Difference | 10 | 16 | 98.64 | 93.70 | **96.10** | **92.51** |
| | D-NeRF | 10 | 16 | 46.34 | 82.07 | 58.41 | 41.42 |
| | C-NERF | 10 | 16 | 92.66 | **99.71** | 96.05 | 92.41 |
| | **Ours** | 10 | **0** | **98.67** | 93.44 | 95.67 | 92.26 |
| Desk | Gaussian Difference | 10 | 10 | 93.99 | 91.88 | 92.78 | 86.75 |
| | D-NeRF | 10 | 10 | 20.81 | 41.72 | 27.74 | 16.42 |
| | C-NERF | 10 | 10 | 96.02 | 90.48 | 93.17 | 87.24 |
| | **Ours** | 10 | **0** | **96.33** | **98.48** | **97.30** | **94.90** |
| Potting | Gaussian Difference | 10 | 11 | 93.54 | 76.51 | 84.31 | 72.67 |
| | D-NeRF | 10 | 11 | 83.87 | 70.77 | 76.71 | 62.23 |
| | C-NERF | 10 | 11 | 71.59 | **94.66** | 81.75 | 69.01 |
| | **Ours** | 10 | **0** | **95.78** | 76.02 | **84.68** | **73.54** |

## A.2 DETAILED RESULTS

Table 6 presents per-scene comparisons for the **NeRFCD** dataset against in-domain baselines. Our method, despite being evaluated in a zero-shot setting, consistently matches or surpasses supervised approaches. On the scenes *Cats*, *Desk*, *Blocks* and *Potting* we achieve state-of-the-art performance, surpassing all other methods. In the remaining *Go pieces* scene we achieve comparable results to the best performing baseline. These results highlight that our zero-shot framework generalizes effectively across scene types without dataset-specific training or auxiliary images.

Table 7 reports per-scene performance on **3DGS-CD** using **F1** and **mIoU** as the primary metrics of comparison. Relative to the 3DGS-CD method, *GOLDILOCS* attains higher *mean* F1 and *mean* mIoU across the five scenes and achieves the top F1/mIoU on the majority of individual scenes. Where the baseline is slightly ahead on a particular scene or single metric, the margin is small.

Table 8 reports results on the **ChangeSim** and **VL-CMU-CD** datasets using the F1 metric. On ChangeSim, our method achieves 52.4%, outperforming some of the existing in-domain methods. On VL-CMU-CD, which is annotated only for removals, our approach again surpasses ZSSCD by over 10% (61.8% vs. 51.6%), although supervised baselines such as C-3PO and SimSaC remain stronger (80.0% and 79.7% respectively). To some extent we attribute this to our higher precision compared to

Table 7: Detailed results per scene on the 3DGS-CD dataset.

| Dataset | Method | # $T_1$ Images Evaluation | Auxiliary↓ | Precision↑ | Recall↑ | F1↑ | IoU↑ |
|---------|--------|------------|------------|------------|---------|-----|------|
| Mustard | 3DGS-CD | 4 | 4 | **97.68** | 98.75 | **98.21** | **96.48** |
|         | **Ours** | 4 | 0 | 96.24 | **99.80** | 97.93 | 96.06 |
| Desk | 3DGS-CD | 4 | 4 | **99.17** | 95.82 | 97.47 | 95.06 |
|      | **Ours** | 4 | 0 | 96.79 | **99.88** | **98.30** | **96.67** |
| Swap | 3DGS-CD | 4 | 4 | **98.84** | **98.82** | **98.83** | **97.69** |
|      | **Ours** | 4 | 0 | 98.72 | 94.35 | 96.20 | 93.12 |
| Bench | 3DGS-CD | 4 | 4 | **96.51** | 96.76 | 96.64 | 93.49 |
|       | **Ours** | 4 | 0 | 95.60 | **99.82** | **97.65** | **95.44** |
| Sill | 3DGS-CD | 4 | 4 | 95.00 | 97.86 | 96.41 | 93.07 |
|      | **Ours** | 4 | 0 | **98.10** | 96.99 | **96.51** | **95.23** |

the ground truth, which may use crude annotation or miss some changes. These results underscore that while in-domain training still provides an edge on specific datasets and annotation protocols, our zero-shot method generalizes better than existing zero-shot approaches without requiring dataset-specific supervision.

Table 9 reports results on the **RC-3D** dataset using the mean average precision (mAP) metric. *GOLDILOCS* achieves a mAP of 0.53 using only RGB input and without any training, outperforming even the RGB-D variant of CYWS-3D. CYWS using only RGB input achieves a mAP of 0.14, while CYWS-3D significantly boosts performance to 0.41, and further improves to 0.50 with RGB-D input. We attribute this performance to the strong geometric priors encoded by the 3D reconstruction model, which enables our method to localize object-level changes with high precision, even in the absence of depth input or dataset-specific training.

Given the unique ground truth format of this dataset, bounding box annotations available on both $I_0$ and $I_1$, we applied the mask propagation procedure in twice, for both viewpoints. Specifically, objects detected in $I_0$ were propagated to $R_{0,1}$, and conversely, objects detected in $R_{0,1}$ were propagated back to $I_0$. This bidirectional strategy enhanced robustness to viewpoint changes and ensured consistent object correspondence across views. The downstream detection and classification steps remained unchanged.

Figure 7 illustrates qualitative results on a scene from the **3RScan** dataset, highlighting moved objects. While *GOLDILOCS* produces 2D change maps, we demonstrate that these outputs can be lifted into 3D space by projecting them onto reconstructed meshes or point clouds. This enables both evaluation on datasets with geometric ground truth and visualization in a 3D context. However, the nature of the dataset introduces several challenges: the scan sequences often include motion blur, out-of-focus frames, occlusions from the person performing the scan, and extremely short camera-to-scene distances. As a result, many frames capture only fragments of objects rather than complete or semantically meaningful views, posing significant difficulties for our method, which relies on at least one image per time point capturing objects in their entirety. Nevertheless, *GOLDILOCS* is able to handle these out-of-distribution inputs and still produce meaningful predictions.

Taken together, these evaluations show that our method is competitive with or superior to strong in-domain baselines in many settings, while surpassing existing zero-shot methods. The framework adapts across synthetic and real domains, binary and multi-class tasks, and both pairwise and multi-view scenarios.

Figure 6 highlights several representative cases where objects have remained in place but undergone non-rigid transformations, such as creasing, denting, squashing and bending. These results demonstrate *GOLDILOCS*'s ability to capture subtle shape deformations that do not involve displacement of the object itself. Note that highly textured surfaces, such as the terrazzo tiles visible in the rug example, can give rise to false positive masks, a limitation of our approach. Such cases underscore

Table 8: Results on the ChangeSim and VL-CMU-CD datasets using the F1 metric.

| Method | Zero-shot / In-domain | ChangeSim | VL-CMU-CD |
|---|---|---|---|
| CSCDNet | In-domain | 32.6 | 71.0 |
| DR-TANet | In-domain | 40.2 | 75.5 |
| SimSaC | In-domain | **69.1** | 79.7 |
| ChangeNet | In-domain | - | 79.0 |
| C-3PO | In-domain | - | **80.0** |
| ZSSCD | Zero-shot | - | 51.6 |
| **Ours** | Zero-shot | **52.4** | **61.8** |

Table 9: Results on the RC-3D dataset using the mean average precision (mAP) metric.

| Method | Training data | Input type | mAP |
|---|---|---|---|
| CYWS | coco-inpainted + KC-3D | **RGB** | 0.14 |
| CYWS-3D | coco-inpainted + KC-3D | **RGB** | 0.41 |
| CYWS-3D | coco-inpainted + KC-3D | RGB-D | 0.50 |
| **Ours** | **Zero-shot** | **RGB** | **0.53** |

the challenge of detecting genuine non-rigid deformations from artifacts and noise introduced by the 3D reconstruction model, as well as illumination phenomena present in the input images.

## A.3 ABLATIONS

To understand the contribution of individual components in our method, we performed ablation studies on the 3D reconstruction components relating to the novel-view synthesis and on the geometric matching module used for multi-view scenarios.

Table 10 reports binary and multi-class IoU metrics with and without 3D stereo reconstruction. Concretely, removing the 3D reconstruction component degrades mean IoU by $\downarrow 9.2$ (27.5%). Per-class IoU drops are: *Added* $\downarrow 4.3$ (17.2%), *Removed* $\downarrow 9.9$ (46.9%), *Moved* $\downarrow 4.9$ (64.5%), *Replaced* $\downarrow 6.1$ (28.6%); *Static* also falls by 20.6 (22.3%). The largest absolute loss is in *Removed*, while the largest relative loss is in *Moved*, indicating that the 3D reconstruction and novel-view synthesis are especially critical for removals and partial relocations.

Figure 9 analyzes robustness to $I_0^j \in \mathcal{I}_r$ selection base on the geometric matching module in the 3DGS-CD dataset. Using lower-ranked reference images leads to a steady decline in F1 and IoU, with reconstruction failure below the 20th percentile due to insufficient correspondences.

Table 11 demonstrates the impact of cross-view voting on multi-image change detection performance across five scenes in the 3DGS-CD dataset. Comparing the results, we observe substantial improvements in all metrics—Precision, F1, and IoU—when cross-view voting is enabled. Notably, while recall remains near-perfect without voting due to the method's high sensitivity (often reaching 100%), the lack of cross-view voting leads to significant drops in precision (e.g., 59.57% vs. 95.60% on Bench, 64.42% vs. 96.79% on Desk), indicating many false positives. Incorporating cross-view voting dramatically reduces these errors, yielding F1 scores exceeding 96% and IoU gains of over 30 points in some cases. This demonstrates that our voting mechanism is critical for filtering noisy or spurious detections and achieving high-precision, object-consistent predictions across views. See Appendix A.5 for detailed overview of the cross-view voting component.

Figure 10 illustrates the correlation between 3D reconstruction performance and multiclass detection metrics, evaluated on 1% of the ChangeSim dataset. As expected, fewer optimization steps result in higher mean loss and lower mIoU. Notably, with just 10 optimization steps, only 5% of the full optimization process, nearly 90% of the final mIoU performance is retained. This highlights the robustness of our method to varying levels of 3D reconstruction quality.

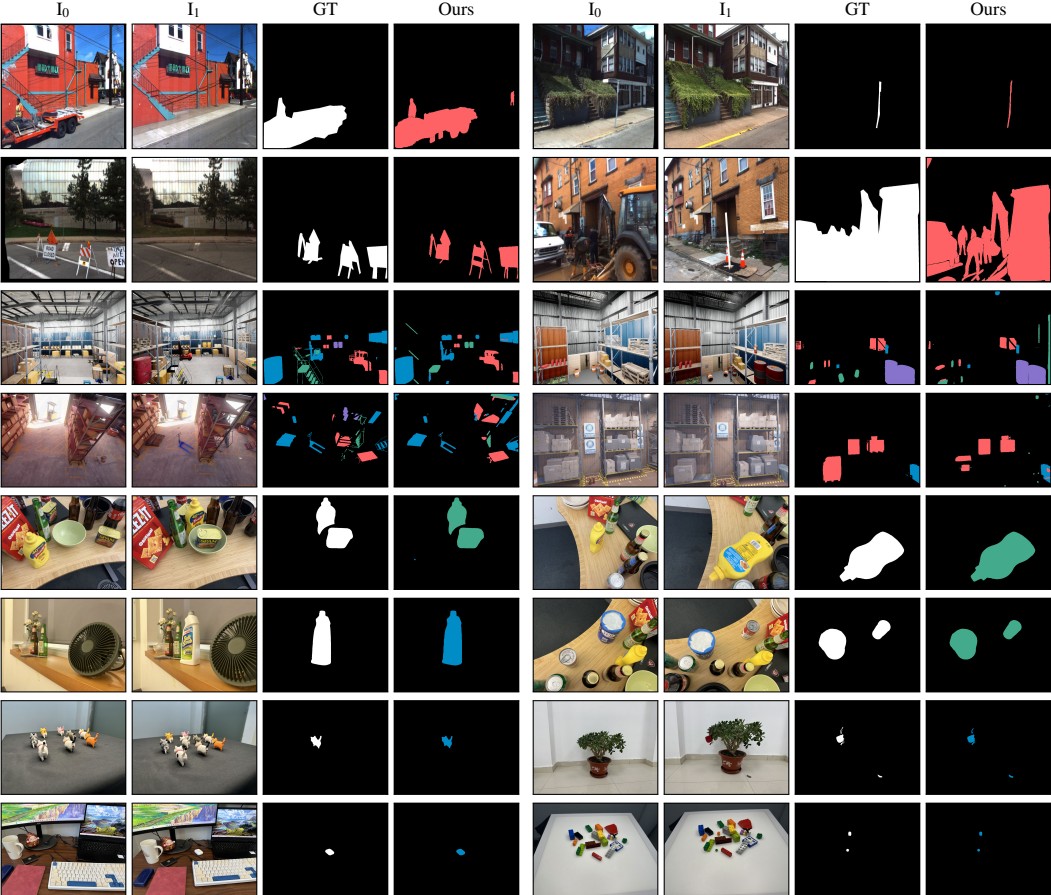

Figure 5: Extended qualitative results of different change types (**Added**, **Removed**, **Moved**, **Replaced**) detected by our method compared to ground truth (GT) in indoor and outdoor scenes. Our results provide color-coded labels even with binary GT and can exceed its accuracy, as annotations are often coarse or miss small changes. This, however, can lower our reported metrics as we compare to imperfect GT.

## A.4 CONFLICT RESOLUTION VIA DEPTH FILTERING

To eliminate geometric inconsistencies and occluders, we clean the pointmaps using a reverse depth test. Let $\mathcal{P}_i$ denote the pointmap reconstructed from image $I_i$, where each entry is a 3D point in world coordinates. For every target view $I_j$, we additionally have a depth map $D_j$ that records the camera-space $z$ value at each pixel, derived directly from the reconstruction process.

Given a world point $p \in \mathcal{P}_i$, we reproject it into the image plane of image $I_j$ to obtain the pixel coordinates $(u, v)$ and its camera-space depth $z_{i \to j}$:

$$(u, v, z_{i \to j}) = C_j p$$

where $C_j$ is the projection operator (intrinsics and extrinsics) mapping world-space points to image coordinates in view $j$.

**Depth Conflict Resolution.** We define the cleaned pointmap $\mathcal{P}_i^{Clean} \subseteq \mathcal{P}_i$ as the subset of points from $\mathcal{P}_i$ that do not conflict with the geometry observed in the opposing view $I_j$. Specifically, a point $p \in \mathcal{P}_i$ is considered *conflicting* with view $j$ if:

$$z_{i \to j} < D_j(u, v), \quad \text{where } (u, v, z_{i \to j}) = C_j p$$

This condition indicates that $p$ would project in front of the surface observed by $I_j$, and therefore represent an occluder or a changed object.

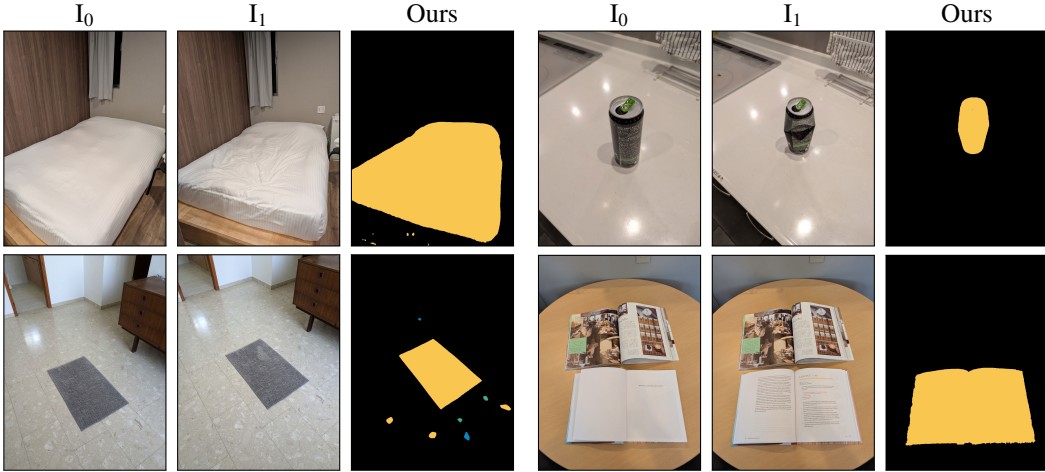

Figure 6: Examples of the Warped change type detection by our method. We present results for changed objects such as creased bedlinen, a dented can, a rug with a foot imprint, and a book with a turned page. Note that highly textured surfaces like the terrazzo tiles in the rug scene pose a challenge for our method, as seen in the false positive masks detected.

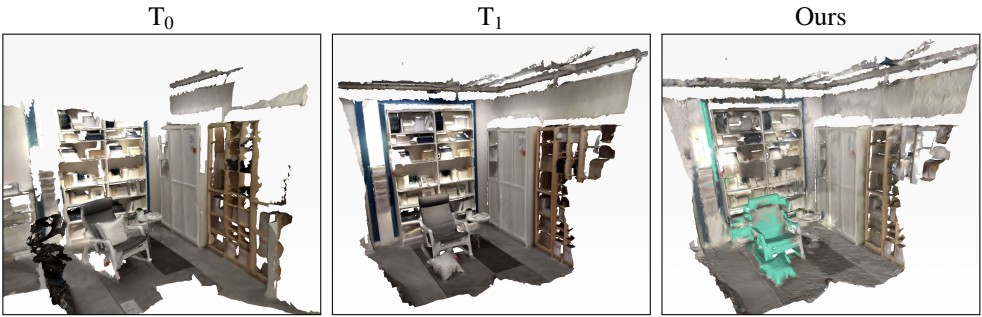

Figure 7: Qualitative results on a scene from the 3RScan dataset. Although *GOLDILOCS* generates 2D change maps, we show that its output can be transformed into a 3D format, enabling broader applicability and evaluation on additional datasets.

By removing geometrically conflicting points through reverse depth filtering, we obtain $\mathcal{P}_0^{Clean}$ and $\mathcal{P}_1^{Clean}$ that exclude points that are likely to correspond to objects that were added, removed, or moved between $T_0$ and $T_1$. The union of these cleaned pointmaps, denoted $\mathcal{P}_*$, is composed of the farthest visible surfaces of the scene and provides a consistent representation of the static scene geometry. A detailed example of this process is illustrated in Figure 8.

**Rendering Pointmaps as Images.** We denote by $R_{i,j}$ the rendered image of a pointmap $\mathcal{P}_i$ from the viewpoint of $I_j$, where each point $p \in \mathcal{P}_i$ is projected using $C_j$ and colored using its appearance in image $I_i$. For example, $R_{0,0}$ denotes a rendering with the geometry and viewpoint of $I_0$, an approximate reconstruction of the original image $I_0$, while $R_{0,1}$ synthesizes how the geometry captured in $I_0$ would appear from the viewpoint of $I_1$, allowing cross-time and cross-view comparisons.

The joint cleaned pointmap $\mathcal{P}_*$ is utilized as a canonical reference geometry for downstream comparisons. For example, to identify objects only visible in $I_0$, we render $\mathcal{P}_*$ using $C_0$, producing $R_{*,0}$. Discrepancies between this rendering and the reference image $I_0$ indicate the presence of dynamic or changed elements. By analyzing such discrepancies in the image space, we can reliably localize and classify the types of changes the objects underwent.

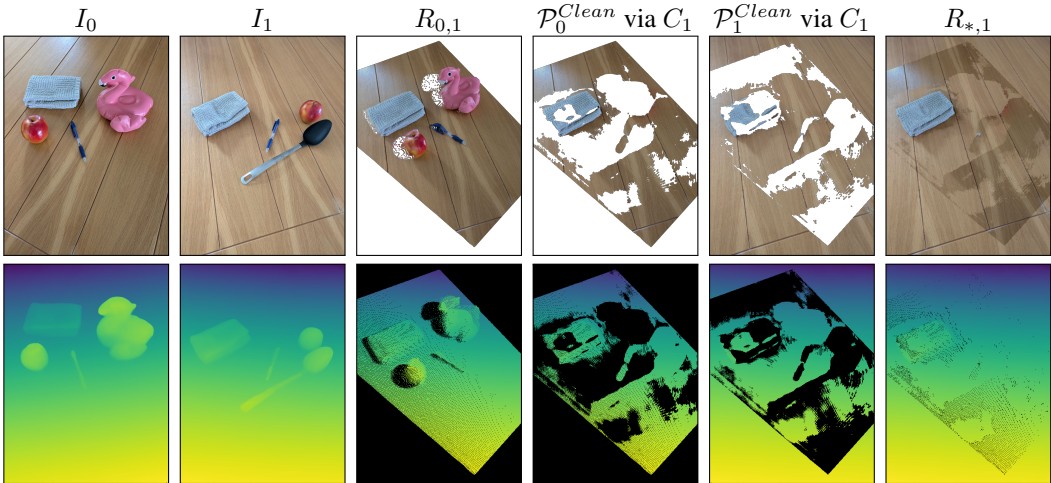

Figure 8: Depth filtering. Estimated depth maps are cleaned by reprojecting 3D points generated from each input image into the viewpoint of the other camera (e.g., $R_{0,1}$, the rendering of $\mathcal{P}_0$ via $C_1$) and identifying geometric conflicts. Points that project in front of corresponding surfaces in the opposite view (e.g., comparing $I_1$ and $R_{0,1}$) are classified as conflicting and removed, yielding cleaned pointmaps, $\mathcal{P}_0^{Clean}$ and $\mathcal{P}_1^{Clean}$. The joint cleaned pointmap $\mathcal{P}_*$ retains only the farthest visible surfaces, corresponding to the persistent, static geometry of the scene. $R_{*,1}$ - the rendering of $\mathcal{P}_*$ from $C_1$ - demonstrates that objects that experienced rigid changes are filtered out, while static yet non-rigidly deformed objects (e.g., the warped tablecloth) remain in the reconstruction.

## A.5  MULTI-VIEW PIPELINE DETAILS

**Pairwise Correspondence Maps.**  For each ordered pair $(I_1^i, I_1^j)$ with $i \neq j$, we compute a per-pixel correspondence map that links each pixel in $I_1^i$ to the pixel in $I_1^j$ observing the same mutually visible 3D point. Let $p \in \mathcal{P}_i$ be the 3D point that projects to $(u, v)$ in $C_i$ and to $(u', v')$ in $C_j$. Collecting all mappings yields a tensor $\text{Corr} \in \mathbb{Z}^{m \times m \times H \times W \times 2}$ defined as:

$$
\text{Corr}[i, j, v, u, :] = \begin{cases} (u', v') & \text{if the 3D point at pixel } (u, v) \in I_1^i \text{ projects inside } I_1^j, \\ (-1, -1) & \text{otherwise.} \end{cases}
$$

Intuitively, $\text{Corr}$ is a per-pixel cross-view map: enabling us to address the relevant pixel —i.e., the same scene point—consistently across all target images.

**Cross-View Voting.**  Let $\hat{I}^i$ be the per-pixel predicted labels for $I_1^i$, where each pixel is labeled from the set $\mathcal{L} = \{Added, Removed, Moved, Warped, Unchanged\}$. Given a segment $s$ in $\hat{I}^i$ whose label is $l \in \mathcal{L}$, let $\mathcal{P}_i(s) \in \mathcal{P}_i$ be the set of 3D points that projects to $s$. We retrieve the coordinates of the pixels of $s$ mapped into $\hat{I}^j$ from $\text{Corr}[i, j, \cdot, \cdot, :]$, denoting the labels of this projected region by $\hat{I}^j(s_{i \to j})$. Each different view $I_1^j \in \mathcal{I}_1$ then votes on the label of $s$ according to the per-pixel majority vote within $\hat{I}^j(s_{i \to j})$:

$$
label_i(s) = l, \qquad label_{j(j \neq i)}(s) = \text{mode}(\hat{I}^j(s_{i \to j}))
$$

If $|\mathcal{I}_1| = m$, then the final label of the segment $s$ is given by cross-view majority:

$$
\widehat{label}(s) = \text{mode}\left(\{label_j(s) \mid j = 1, \ldots, m\}\right).
$$

Ties are resolved by a fixed priority over $\mathcal{L}$. This ensures that each segment casts exactly one vote per view towards the final prediction, leading to consistent labeling across all target images.

Finally, for completeness we include an overview of the multi-view pipeline used in our method (Figure 11). This schematic illustrates the data flow from unaligned multi-view inputs through 3D

Table 10: Ablation results (IoU) on 10% of the ChangeSim dataset.

| | Binary | | | Multi-class | | | | | |
|---|---|---|---|---|---|---|---|---|---|
| Method | Change | Static | mIoU | Added | Removed | Moved | Replaced | Static | mIoU |
| Ours w/o 3D reconstruction | 20.7 | 71.9 | 46.3 | 20.7 | 11.2 | 2.7 | 15.2 | 71.9 | 24.3 |
| Ours | **37.5** | **92.5** | **65.0** | **25.0** | **21.1** | **7.6** | **21.3** | **92.5** | **33.5** |

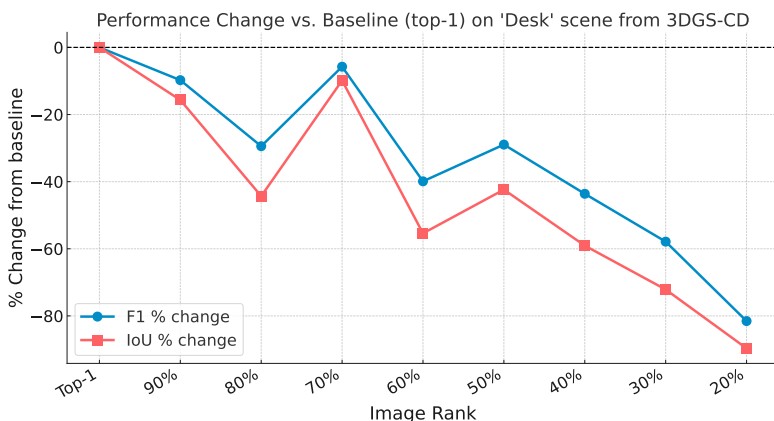

Figure 9: Performance change in F1 and IoU compared to the top-1 baseline on the *Desk* scene from the 3DGS-CD dataset. Below the 20th percentile, reconstruction fails due to insufficient matches.

reconstruction, correspondence filtering, and mask propagation, providing context for the datasets and results reported above.

## A.6 PARAMETERS

Table 12 presents the values of the parameters used for SAM2 in the segmentation step, as well as the $\alpha$ parameter utilized in the filtering step. Although the method exposes several tunable parameters, most settings are shared across all datasets and experiments; we chose values derived from our insights regarding dataset characteristics (image resolution, scene depth, typical pixel size of targets, and appearance ambiguity). Although we believe a thorough parameter search can further enhance results on specific data (or even specific examples), we refrained from doing this.

**SAM2.** The parameters control how the automatic mask generator samples prompts and filters candidate masks. Specifically, they determine (i) the spatial and multi-scale sampling used to propose masks, (ii) the quality thresholds that accept or reject proposals, and (iii) options that broaden or refine the set of candidates. *points_per_side* sets the density of a regular grid of point prompts over the image; larger values probe the scene more densely and typically improve recall at higher compute cost. If *crop_n_layers* $> 0$, SAM2 also runs on an image-pyramid of crops to better capture small objects; *crop_n_points_downscale_factor* reduces the grid density on deeper crop levels to keep runtime manageable. After proposing masks, two quality gates prune weak results: *pred_iou_thresh* keeps only masks whose *predicted IoU* (the model's internal confidence in mask quality) exceeds the threshold, and *stability_score_thresh* keeps masks that remain stable under small binarization perturbations. *multi_mask_output* returns multiple plausible masks per prompt, useful for instances where boundaries are ambiguous. *use_m2m* is a SAM2 flag that enables mask-to-mask refinement, where previously predicted masks are used as prompts to refine final mask prediction.

**Filtering.** To prevent false positives in occluded or out-of-view regions, we apply a visibility-aware filtering step to all tracked masks. For each predicted mask, we compute the fraction of its area that lies inside $R_{0,1}$ and keep the mask only if this fraction is at least $\alpha$. This suppresses detections that are supported mainly by pixels that are occluded or outside the shared field of view. Datasets with strong inter-view overlap and larger scene depth (e.g., *ChangeSim*) can use a higher $\alpha$, since

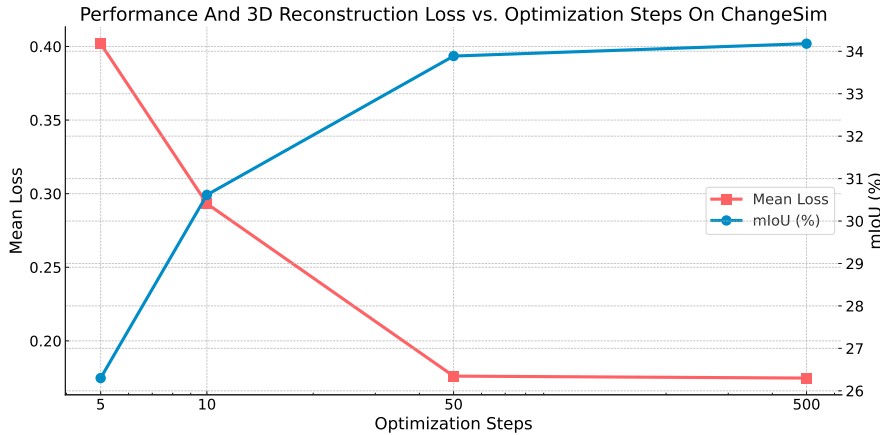

Figure 10: Relationship between 3D reconstruction loss and multiclass mIoU as a function of optimization steps. A clear correlation is observed: as mean loss decreases, performance (mIoU) improves. Notably, nearly 90% of the final performance is achieved with just 5% (10 steps) of the total optimization process.

Table 11: Ablation results of the cross-view voting component on the 3DGS-CD dataset.

| Dataset | Method | Precision↑ | Recall↑ | F1↑ | IoU↑ |
|---------|--------|-----------|---------|-----|------|
| Bench | Ours w/o cross-view voting | 59.57 | **100.00** | 73.48 | 59.57 |
| | **Ours** | **95.60** | 99.82 | **97.65** | **95.44** |
| Desk | Ours w/o cross-view voting | 64.42 | **100.00** | 76.98 | 64.42 |
| | **Ours** | **96.79** | 99.88 | **98.30** | **96.67** |
| Mustard | Ours w/o cross-view voting | 72.80 | **99.99** | 83.38 | 72.79 |
| | **Ours** | **96.24** | 99.80 | **97.93** | **96.06** |
| Sill | Ours w/o cross-view voting | 66.97 | **97.19** | 77.90 | 66.42 |
| | **Ours** | **98.10** | 96.99 | **96.51** | **95.23** |
| Swap | Ours w/o cross-view voting | 79.93 | **95.13** | 86.56 | 78.06 |
| | **Ours** | **98.72** | 94.35 | **96.20** | **93.12** |

most objects are mutually visible and less affected by occlusion or parallax. Datasets with sparser overlap and smaller scene depth (e.g., *3DGS-CD*) benefit from a lower $\alpha$, because objects may be only partially observed in each image and are more susceptible to occlusions and parallax; relaxing the filter avoids discarding such cases.

**Parameter selection.** We selected the SAM2 parameters *per dataset* based on four practical factors: the input image resolution, the expected scene depth (and resulting parallax/occlusions), the typical *pixel scale* of target objects, and their appearance ambiguity (e.g., objects sharing color/texture with nearby surfaces). These considerations drove the balance between coverage and precision (grid/crops vs. thresholds) and when to allow multiple candidates per prompt.

For **ChangeSim**, we enable *multi_mask_output*, use a dense grid (64) and one crop layer, and set moderate quality gates (`pred_iou_thresh = stability_score_thresh = 0.8`). The denser sampling supports smaller targets at the given resolution, while the relaxed thresholds tolerate appearance ambiguity (e.g., a box with nearly the same color as its shelf), which would otherwise be over-pruned. **3DGS-CD** and **NeRFCD** (indoor multi-view datasets) retain the same dense grid and single crop to preserve recall at their object scales, but raise both gates to $0.9/0.9$ to improve precision

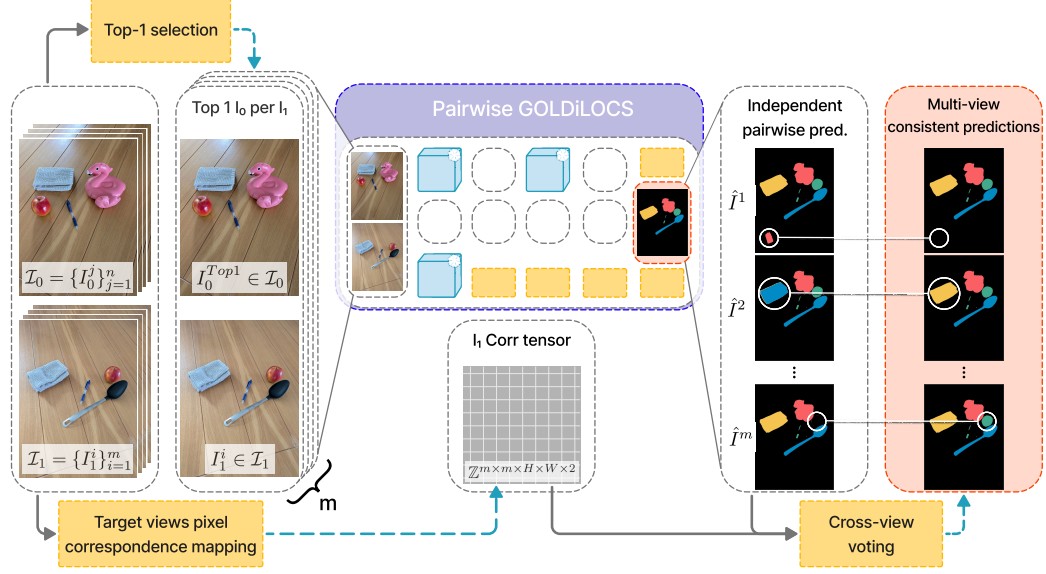

Figure 11: Overview of the multi-view pipeline.

Table 12: Parameters used for SAM2 and $\alpha$ values used in GOLDILOCS's filtering step.

| Dataset | Filtering $\alpha$ | SAM2 pred_iou_thresh | stability_score_thresh | multi_mask_output | use_m2m | points_per_side | crop_n_layers | crop_n_points_downscale_factor |
|---|---|---|---|---|---|---|---|---|
| ChangeSim | 0.8 | 0.8 | 0.8 | True | False | 64 | 1 | 2 |
| 3DGS-CD | 0.25 | 0.9 | 0.9 | True | False | 64 | 1 | 2 |
| NeRFCD | 0.25 | 0.9 | 0.9 | True | False | 64 | 1 | 2 |
| VL-CMU-CD | - | 0.8 | 0.8 | False | False | 32 | 0 | 0 |

under clutter and depth-induced inconsistencies. Finally, **VL-CMU-CD** disables multi-mask output, uses a lighter grid (32) and no crops, with mid-strict gates ($0.8/0.8$) in order to avoid detection of smaller objects not annotated by the creators of the dataset. In summary, higher thresholds address precision when depth/clutter increase false positives; denser grids and crop layers recover small objects in higher-resolution settings; and enabling multi-mask helps when class/texture similarity makes boundaries ambiguous. See Table 13 for an analysis of how different detection thresholds affect performance on the VL-CMU-CD dataset.

**3D reconstruction.** We selected MASt3R parameters that faciliate accurate reconstruction with an emphasis on rendering accurately aligned 2D images. In order to be able to infer on each pixel of the input data it was important to avoid pointmap cleaning at the reconstruction phase as we aimed to retain all of the scene geometry in both time points, such that we can clean the pointmaps with our depth filtering logic. Below are the parameters used for 3D reconstruction:

- `lr1` = 0.07
- `niter1` = 500
- `lr2` = 0.014
- `niter2` = 200
- `optim_level` = 'coarse'
- `matching_conf_thr` = 0
- `shared_intrinsics` = False
- `desc_conf` = '3d'
- `scenegraph_type` = 'complete'
- `winsize` = 1

Table 13: Analysis of SAM2 parameter settings on VL-CMU-CD and their impact on binary change detection performance. Following prior work, the primary metric for the VL-CMU-CD dataset is $F1_{fg}$. All other SAM2 and filtering parameters remain as presented in Table 12.

| Dataset | SAM2 | | Binary Metrics (%) | | | | | | |
|---|---|---|---|---|---|---|---|---|---|
| | pred_iou_thresh | stability_score_thresh | $F1_{avg}$ | $F1_{fg}$ | $F1_{bg}$ | $Prec_{fg}$ | $Recall_{fg}$ | $Prec_{bg}$ | $Recall_{bg}$ |
| VL-CMU-CD | 0.7 | 0.7 | 78.3 | 59.5 | 96.9 | 53.6 | **79.4** | **98.2** | 96.1 |
| VL-CMU-CD | 0.8 | 0.8 | **79.6** | **61.8** | 97.4 | 60.5 | 72.7 | 97.7 | 97.4 |
| VL-CMU-CD | 0.9 | 0.9 | 72.4 | 47.1 | **97.7** | **62.8** | 43.7 | 96.2 | **99.4** |

- `win_cyclic` = False
- `refid` = 0
- `min_conf_thr` = 0.0
- `tol` = 0.0
- `cam_size` = 0.2
- `TSDF_thresh` = 0.0
- `as_pointcloud` = True
- `mask_sky` = False
- `clean_depth` = False
- `clean_with_conf` = False
- `transparent_cams` = False

## A.7 LARGE LANGUAGE MODELS (LLMS) USAGE

In this paper we utilized LLMs as writing support tools, limited to rephrasing sentences, shortening paragraphs and refining wording in select instances. Note that LLMs were not used for ideation, literature discovery, paper summarization, nor the methodology development.

