# OpenReview forum: "GOLDILOCS: GENERAL OBJECT-LEVEL DETECTION AND LABELING OF CHANGES IN SCENES"
_ICLR.cc/2026/Conference — ICLR 2026 Poster_

### Official Review · Reviewer_14W9 · 2025-10-31

**Soundness:** 3
**Presentation:** 3
**Contribution:** 3
**Rating:** 6
**Confidence:** 3

**Summary:**

This paper proposes GOLDILOCS, a zero-shot, 3D-aware framework for object-level scene change detection (SCD) from image pairs or multi-view image sets. The method reformulates SCD as a combination of stereo 3D reconstruction, semantic segmentation, mask tracking, and non-rigid change detection via SSIM-based comparison. It leverages foundational models (MASt3R for 3D reconstruction and SAM2 for segmentation) without additional training and introduces a standardized object-level change taxonomy including Removed, Added, Moved, and Warped. Experiments on both synthetic and real datasets (ChangeSim, VL-CMU-CD, 3DGS-CD, NeRFCD) demonstrate strong zero-shot performance, especially in IoU and F1 metrics, while avoiding expensive multi-view reconstruction at test time.

**Strengths:**

1. Zero-shot object-level change detection with 3D geometric reasoning.

2. Modular pipeline leveraging stereo reconstruction and segmentation models.

3. Introduces a clear object-level change taxonomy (Removed, Added, Moved, Warped).

4. Faster inference compared to traditional 3D reconstruction pipelines.

5. Comprehensive experiments across multiple datasets.

**Weaknesses:**

Ⅰ. Parameter Sensitivity:

SSIM Thresholds: The thresholds used for non-rigid change detection significantly affect results. A threshold set too high may miss subtle deformations, while a threshold too low may falsely classify static objects as changed.

Label Priority Ordering: The per-pixel prediction relies on the priority order (Warped > Moved > Removed > Added). This can affect nested or overlapping object changes; for example, small objects inside a larger moved object may be suppressed and misclassified.

Ⅱ. Limited Ablation:

Ablation experiments mainly focus on stereo reconstruction and novel view synthesis, but the contribution of non-rigid change detection and mask propagation is not thoroughly analyzed.

The effects of individual components such as depth filtering, cross-view voting, and mask propagation on each change type (added, removed, moved, warped) are not quantified.

Ⅲ. Data Limitations:

The evaluation datasets lack extreme real-world scenarios, including large occlusions, fast-moving objects, or highly dynamic scenes, which may cause 3D reconstruction or mask tracking failures.

**Questions:**

1. How sensitive is GOLDILOCS to SSIM thresholds and mask tracking parameters across datasets?

2. How does it handle partially occluded or highly dynamic objects?

3. Could hierarchical segmentation improve detection for nested objects?

4. Would longer temporal sequences improve detection robustness?

5. Any plans for evaluation in real-world dynamic scenes?

---

> ### Author Response · Authors · 2025-11-21
>
> Reviewer 14W9
>
> Thank you for the insightful comments. We address each point in detail below.
> For your convenience added sections in the revision appear in blue.
>
> "Ⅰ. Parameter Sensitivity ..."
>
> We appreciate this observation and added it to our limitations (section 7).
> Our method uses a data-driven statistical threshold, as described in section 4.4:
>
> “We calculate the mean dissimilarity score of each static segmentation mask in MI₁, and classify those farther than one standard deviation from the mean across all masks as warped.”
>
> This approach adapts to the distribution of each scene rather than relying on a fixed value, which we found to work robustly across datasets. While this strategy is generally stable, more sophisticated techniques may further improve edge cases.
>
> “Label Priority Ordering ...”
>
> We would like to refer the reviewers and point that we explicitly acknowledge this in our limitations (section 7). Importantly, such cases are rare and highly task-specific, occurring only when (1) evaluating multiclass SCD datasets, which are far less common than binary datasets, and (2) nested or overlapping objects exist in the scene with sufficiently complex ground-truth definitions.
> In ChangeSim—the only multiclass dataset evaluated—we did not encounter examples where the priority ordering caused misclassification.
>
> "Ⅱ. Limited Ablation ..."
>
> We would like to highlight that non-rigid change detection is not a pipeline component, it is a prediction task, and as such it cannot be ablated. Furthermore, the non-rigid prediction process operates only after rigid changes are identified, hence it does not influence any other change types; it merely flags warped objects among those classified as static.
>
> Regarding cross-view voting: this mechanism is only applicable to image-set datasets (many-to-many), and the datasets of this type we evaluate on provide binary annotations only. This means that per-category effect analysis is impossible because ground truth for added/removed/moved/warped is not available. As per the reviewer suggestion we added a binary-only ablation of cross-view voting for the 3DGS-CD dataset, as seen in the newly added Table 11 available in the new revision.
>
> Regarding mask propagation and depth filtering: these are core structural components, not optional modules. Without mask propagation, the method cannot determine object correspondences between t₀ and t₁, and thus cannot detect changes at all. Without depth filtering, the method cannot perform the 3D conflict resolution that enables object-level detection.
>
> These components are therefore not ablatable without disabling the method entirely.

---

> ### Author Response · Authors · 2025-11-21
>
> "Ⅲ. Data Limitations ..."
>
> We appreciate the opportunity to clarify our evaluation choices. We selected datasets that are widely used benchmarks in the SCD community, enabling comparisons with the largest possible set of related work. As outlined in Table 1, these datasets include real-world indoor and outdoor scenes subject to illumination changes, weather variation, seasonal shifts, occlusions, and dynamic elements such as vehicles, animals, and people.
>
> In this sense, the scenes are very dynamic - seasonal changes can alter the view dramatically, and objects can undergo a variety of rigid and non-rigid changes between t₀ and t₁. If the question refers to objects moving during the moment of capturing the image, resulting in motion blur for example, we consider that to be an artifact, as we assume that at each time point the image captures a scene that is not in the process of changing, but either before (t₀) or after (t₁) the change. Heavy motion blur is an artifact that can break both reconstruction and segmentation which is not unique to our method, and we explicitly added this point to the limitations.
>
> Large occlusions are present in both image-pair and image-set datasets.
> See Figure 3 (second row, left) and Figure 5 (third row, right) for examples where our method successfully detects occluded changed objects.
>
> “How sensitive ...”
>
> Our SSIM-based non-rigid detection uses a consistent statistical rule across all experiments and datasets. Note that non-rigid changes are not annotated in any existing dataset we evaluate, and to our knowledge, we are the first RGB-based SCD method to detect object-level non-rigid deformations.
>
> SAM2 does not expose mask tracking parameters, but per the reviewers suggestions we performed an analysis regarding the detection thresholds of SAM2 on the VL-CMU-CD dataset, which can be found in the newly added Table 13 in the revision. We would like to refer to Appendix A.5 and Table 12 regarding discussion of parameters, which were held nearly constant across datasets.
>
> “Could hierarchical segmentation ...”
>
> Potentially, yes. We appreciate this observation and will add it to the future work section.
>
> “Would longer temporal sequences ...”
>
> Our method’s input is either a pair of images (t₀, t₁) or two image sets capturing the states of a scene at t₀ and t₁. There is no temporal dependency beyond this difference. If the question refers to denser image sets at t₀ for multi-image scenes, then more coverage can help reconstruction quality and promote better matches between images of different timestamps, but all images in a set are considered to be taken at the same time.

---

> > ### Comment · Reviewer_14W9 · 2025-11-25
> > **Assessment of the Author Response**
> >
> > Thank you for the detailed and well-structured rebuttal. Your clarifications address many of the concerns raised in the initial review, and I appreciate the additional analyses and revisions you have incorporated.
> > Regarding parameter sensitivity, the explanation of a data-driven statistical threshold helps clarify how the method adapts across datasets. While some edge-case sensitivity may remain, the revised discussion in the limitations section is satisfactory.
> > On the issue of label priority and nested objects, I acknowledge your clarification that such cases are rare in existing multiclass datasets and that ChangeSim did not exhibit such failures. The explicit discussion of this limitation in the paper is appropriate.
> > Concerning the ablation analysis, your explanation that non-rigid detection is not an ablatable module is reasonable, and the additional binary ablation for cross-view voting strengthens the empirical section. The clarification that mask propagation and depth filtering are structural requirements also resolves the feasibility concern.
> > For data limitations, your justification for dataset selection and the added examples illustrating robustness under occlusions are appreciated. The added discussion about motion blur artifacts is appropriate. I also appreciate the new parameter analysis for SAM2 and the consistency of your SSIM-based procedure across datasets.
> > Your acknowledgement of the potential benefits of hierarchical segmentation and denser image sets is helpful, and it is good to see these points reflected in the revised manuscript and future work section.
> > Overall, the rebuttal satisfactorily addresses most of my concerns. While some broader evaluations—particularly on more challenging dynamic real-world scenes—remain an open direction, the clarifications and additions have strengthened the work.
> > Thank you for the careful revisions.

---

### Official Review · Reviewer_x35Y · 2025-11-01

**Soundness:** 3
**Presentation:** 3
**Contribution:** 2
**Rating:** 4
**Confidence:** 4

**Summary:**

The paper introduces GOLDILOCS, a zero-shot and pose-agnostic approach for object-level semantic change detection (SCD). The key idea is utilizing  3D geometric inconsistencies to detect changes. With the estimated pointmaps by dense stereo reconstruction models , the method generates reference images of clean point clouds. Changes are then identified via mask tracking for rigid objects and SSIM heatmaps for non-rigid ones. Evaluations on various datasets show better performance over baseline methods.

**Strengths:**

- With the help of SAM2, the proposed method is zero-shot.
- The idea to utilize SAM and MASt3R to SCD problem is practical.
- The proposed method achieved best overall performance in most settings.

**Weaknesses:**

- The proposed method depends on two large foundation models, which demands large computations and may limit its applications.
- Lack of implementation details/runtime comparisons to show the extra cost of the proposed pipeline.
- The peromance improvement is marginal (e.g, resutls in Tab.4.)

**Questions:**

- Is there analysis of how the performance/generality of 3D reconstruction model affect the result of SCD results? Are there any visualization of reconstructed results?
- The “conflict resolution” in Sec. 4.2 is confusing, what does it mean?
- Is the method end-to-end? Or they are rule-based after getting the segmentation/reconstruction by existing models.
- Citation of 3DGS-CD of Tab.1 differs from that of Tab. 2.

---

> ### Author Response · Authors · 2025-11-21
>
> Reviewer x35Y
> We thank the reviewer for the thoughtful evaluation. Below, we address each point in detail.
> For your convenience added sections in the revision appear in blue.
>
> “The proposed method ... may limit its applications ... Lack of implementation details/runtime ...”
>
> We respectfully disagree with this characterization. As shown in the newly added Table 5 available in the new revision, our method is 10–100× faster than prior work when measured on multi-image scenes, and all methods were evaluated on similar or identical hardware (NVIDIA RTX 3090).
> Additionally, our method is zero-shot—requiring no model training, no dataset-specific tuning, and no labeled data, which drastically reduces end-to-end system cost. Competing methods rely on pretraining, fine-tuning, and large volumes of auxiliary data, all of which are entirely avoided in our pipeline.
> Regarding runtime: for a single image pair, the pipeline runs in under 30 seconds, using <8GB GPU RAM, depending on resolution and filtering threshold. Multi-image scenes can be completed in under 10 minutes, compared to several hours (or days) for baselines like C-NeRF or Gaussian Difference.
> Applications of SCD include tasks such as environmental monitoring, legal surveillance, and map updating are inherently offline and asynchronous, where accuracy and generalization matter more than runtime, and as such, are generally not constrained by computation resources.
>
> “The performance ...”
>
> We respectfully disagree with this assessment. When evaluated under equal conditions—as in the ChangeSim dataset (Table 3) our method achieves a substantial performance gain: +20.8% in multiclass mIoU and +8.9% in binary mIoU over the closest competitor, which is itself supervised and evaluated in-domain, while our method operates in a fully zero-shot setting. In contrast, the results in Table 4 reflect inherently biased conditions that favor existing methods: those approaches require 4 to 100 auxiliary images, with inference times ranging from several hours to over a day. Our method, by comparison, operates without any auxiliary inputs and produces results in minutes, as seen in the newly added Table 5 available in the new revision. The fact we still outperform these methods despite the asymmetry in resource requirements only underscores the strength of our approach.
>
> “Is there analysis ... performance/generality of 3D reconstruction ...”
>
> We address this in Table 10, which provides an ablation of the reconstruction module and its impact on downstream SCD performance. Additionally as per the reviewers suggestion, we conducted an experiment exploring the relationship between 3D reconstruction optimization steps and reconstruction loss, serving as a proxy for performance and generality, and the multiclass mIoU metric on the ChangeSim dataset. The results of this experiment are shown in the newly added Figure 8 available in the new revision.
>
> We will include qualitative visualizations of pointmaps in the appendix for illustration.
>
> “The ‘conflict resolution’ ...”
>
> Thank you for the suggestion—we agree that this section can benefit from visual support. We will include a figure in the revised version to clarify the reverse depth filtering process. This process is core to our approach and will be made more transparent with visual aid.
>
> “Is the method ...”
>
> We would like to refer the reviewer and point that the logic following the reconstruction process and segmentation is described in 4.3, including multiclass prediction logic and the filtering step.
>
> “Citation of 3DGS-CD of Tab.1 differs from that of Tab. 2.”
>
> Thank you for spotting this, we have corrected this issue.

---

> > ### Author Response · Authors · 2025-11-25
> >
> > Dear Reviewer x35Y,
> >
> > As per your following insightful suggestions:
> > “Is there analysis ... performance/generality of 3D reconstruction ...”
> > “The ‘conflict resolution’ ...”
> >
> > We have included a new Figure 7 in the revised manuscript, illustrating the depth-filtering process with renderings and qualitative visualizations of both intermediate and final outputs from the 3D reconstruction module, in aim to aid the reader’s understanding of this stage of our pipeline.
> >
> > Please note that due to the addition of Figure 7 to the manuscript, the results of our previously added experiment exploring the relationship between 3D reconstruction performance and detection performance on the ChangeSim dataset are now available in Figure 9, instead of Figure 8.
> >
> > Thank you very much for your time,
> > The Authors.

---

### Official Review · Reviewer_rCUL · 2025-11-04

**Soundness:** 2
**Presentation:** 2
**Contribution:** 2
**Rating:** 6
**Confidence:** 3

**Summary:**

The authors introduce GOLDILOCS, a framework that establishes a standardized taxonomy for object-level change labeling in 3D scenes. The approach is grounded in geometric reconstruction and visibility reasoning, where an object is defined as any visually identifiable entity ranging from volumetric elements like boxes or furniture to planar textures such as ink stains on paper.

GOLDILOCS reconstructs the 3D geometry of a scene and employs novel-view rendering, segmentation, and temporal tracking of object masks to detect both rigid and  non-rigid transformations over time. The framework is evaluated across multiple datasets, demonstrating consistent and satisfactory improvements over state-of-the-art methods in object-level change detection and labeling.

**Strengths:**

The paper introduces a well-defined and systematic taxonomy for object-level change labeling in 3D scenes, addressing the lack of standardization in existing approaches.

The authors adopt an inclusive definition of object, covering both volumetric entities (e.g., furniture) and planar or texture-based elements (e.g., ink blotches), which enhances the framework’s generality and applicability.

The integration of 3D geometric reconstruction and visibility-based reasoning allows for more precise understanding of scene changes compared to purely 2D or image-based methods.

The use of segmentation and mask tracking across time ensures consistent identification of objects and their transformations, including non-rigid changes.

The pipeline’s inclusion of novel-view rendering improves the robustness of change detection under different viewpoints and occlusions.

The model has been tested on multiple datasets, demonstrating strong generalization and satisfactory improvements over state-of-the-art methods.

By combining geometry, visibility, and temporal reasoning, GOLDILOCS can be extended to practical domains like robotics, AR/VR scene updates, and autonomous navigation.

**Weaknesses:**

The proposed framework is computationally heavy, as it integrates conventional segmentation and 3D stereo reconstruction modules, making it less efficient for real-time or large-scale applications.
The 3D stereo reconstruction component appears to rely on existing methods with minimal innovation, reducing the novelty of that part of the pipeline.
The paper lacks clear technical details about the underlying 3D reconstruction model its architecture, parameters, and optimization strategy are not adequately discussed.
The motivation for selecting specific existing models for segmentation and reconstruction is not well justified. The rationale behind these design choices should have been elaborated to strengthen the methodological clarity.
The computational complexity, including FLOPs, memory requirements, and inference time, is not reported. Such analysis would be valuable for early reference and comparative evaluation.
Given its reliance on multiple integrated modules, the pipeline may face scalability and deployment challenges in dynamic or resource-limited environments.

**Questions:**

The proposed framework is computationally heavy, as it integrates conventional segmentation and 3D stereo reconstruction modules, making it less efficient for real-time or large-scale applications.
The 3D stereo reconstruction component appears to rely on existing methods with minimal innovation, reducing the novelty of that part of the pipeline.
The paper lacks clear technical details about the underlying 3D reconstruction model its architecture, parameters, and optimization strategy are not adequately discussed.
The motivation for selecting specific existing models for segmentation and reconstruction is not well justified. The rationale behind these design choices should have been elaborated to strengthen the methodological clarity.
The computational complexity, including FLOPs, memory requirements, and inference time, is not reported. Such analysis would be valuable for early reference and comparative evaluation.
Given its reliance on multiple integrated modules, the pipeline may face scalability and deployment challenges in dynamic or resource-limited environments.

---

> ### Author Response · Authors · 2025-11-21
>
> Reviewer rCUL
>
> We appreciate the reviewer’s constructive feedback and address the concerns point-by-point below.
>  For your convenience added sections in the revision appear in blue.
>
> “The proposed framework ... less efficient for real-time ...”
>
> We respectfully disagree with the assertion that our method is computationally heavy or impractical for deployment. On the contrary, as shown in the newly added Table 5 available in the new revision, our method is 10–100× faster than existing approaches when measured on multi-image scenes, and runs on similar or identical hardware (e.g., NVIDIA RTX 3090). Importantly, our pipeline is zero-shot, meaning it requires no training, no data labeling, and no fine-tuning, which significantly reduces total computational cost compared to supervised baselines.
>
> While it's true that our method integrates segmentation and stereo reconstruction, this is done in a modular and highly efficient way. For image pairs, the full pipeline runs in under 30 seconds, using less than 8GB of GPU memory, depending on resolution and detection thresholds. Multi-image scenes can be processed in under 10 minutes, compared to hours or days for baselines like C-NeRF and Gaussian Difference.
>
> Moreover, the assumption that real-time performance is required is misguided in the context of scene change detection (SCD). By definition, SCD involves detecting changes between two temporally separated observations (t₀ and t₁). This is not a live inference task. Applications such as environmental monitoring, legal surveillance, and map updating are inherently offline and asynchronous, where accuracy and generalization matter more than runtime.
>
> The core point remains: our method is among the most computationally efficient SCD pipelines currently available, especially considering it achieves state-of-the-art results under zero-shot conditions.
>
> “The 3D stereo reconstruction ... parameters, and optimization ...”
>
> We believe this comment reflects a misunderstanding of the role that the 3D reconstruction module plays in our framework. Our contribution is not to propose a new reconstruction algorithm, but to redefine how reconstruction is used within the scene change detection task.
>
> To emphasize our contributions:
>
> We are the first to reframe image-based SCD as 3D stereo reconstruction across time, using 3D volumetric conflict to identify object-level changes in a zero-shot setting.This is a conceptual and practical innovation not found in prior work.
> First method that can perform binary and multiclass change detection under zero-shot conditions on both image pairs and sets, achieving state of the art results across all evaluated datasets.
> We are the first method to detect non-rigid changes from unaligned RGB image pairs under zero-shot conditions.
> Our method is 10x faster compared to related work when measured on multi-image scenes.
>
> The 3D module we use (MAST3R) is treated as a frozen component. Our pipeline does not compare full scene reconstructions or rely on model-to-render alignment. Instead, we feed as little as a single image pair into a reconstruction model and extract fine-grained semantic changes across time in a fully zero-shot manner.
>
> We have include parameter details and considerations for MAST3R in the appendix section A.5, though we stress that our methodological contribution lies in how reconstruction is used, not how it is computed.

---

> ### Author Response · Authors · 2025-11-21
>
> “The motivation for selecting specific existing models ...”
>
> We appreciate the opportunity to clarify our design choices.
>
> We selected MAST3R for 3D stereo reconstruction due to its state-of-the-art performance on uncalibrated image inputs and its ability to produce dense geometry and camera matrices—outputs critical to our 3D reasoning framework. Importantly, MAST3R is used as a modular component; it can be replaced by any alternative that provides comparable geometric outputs.
>
> For segmentation, we use SAM2, which offers class-agnostic, state-of-the-art instance segmentation and propagation. Again, this component is modular—any mask-generating and tracking system can be substituted.
> These choices were made deliberately to maximize generalization and performance, as the pipeline does not depend on any one reconstruction or segmentation model. We have clarified this rationale in the revised manuscript, see section 4.

---

### Official Review · Reviewer_KMk1 · 2025-11-05

**Soundness:** 3
**Presentation:** 3
**Contribution:** 2
**Rating:** 4
**Confidence:** 4

**Summary:**

The paper presents a new method for scene change detection. The new method, entitled GODILOCS, offers an alternative for zero-shot, class-agnostic object-level change detection, leveraging recent advances in foundation models. The method is 3D geometry aware, firstly by calculating the 3D geometry of the scene using Mast3R. The created point-maps are filtered for inconsistent regions (likely corresponding to changes in the scene) and then images are rendered using the same viewpoints such that the images can be compared. Discrepancies between the rendered and initial images help identify object-level changes, with the help of object-level masks created by SAM2. A foundation model for tracking is the afterwards deployed to differentiate between rigid and non-rigid changes. The presented method achieves state-of-the-art results compared with a set of relevant baselines both on synthetic and real-world dataset.

**Strengths:**

The paper addresses an interesting topic: scene change detection. Especially, it also helps sub-categorize the change into added, removed, or moved objects (rigid and non-rigid changes). Scene change detection is an understudied and interesting topic.

The paper leverages the latest trends in foundation models, offering a class-agnostic, zero-shot method for scene change detection. More specifically, using Mast3R, the proposed method creates a 3D reconstruction of the scene to have a 3D geometry-aware 3D change detection, and then uses a class-agnostic instance segmentation model (SAM2) to identify objects in the scene and track them using DEVA].

The paper achieves SoTA results, compared to other baselines, validating the motivation of having a 3D geometry aware change detection methods when just images of a scene are given.

The paper is well-motivated, and well-written. The methodology is thoroughly explained.

**Weaknesses:**

The novelty of the paper is limited. The paper is mainly extending the work of [1] that uses Mast3R towards creating a geometry aware-scene change detection method. Mast3r offers the 3D reconstruction of the scene, along with the poses and the calibration matrices of the views. Through that, the proposed method can then compare depth maps and render novel views, using the same viewpoints. On the rendered views, the methodologies applied in [1] are then deployed.

More baselines should be included, including CYWS-3D (Sachdeva & Zisserman, 2023b). Even though the paper offers bounding boxes as changing regions, SAM could be deployed to get the mask for the bounding boxes. Moreover, since the method does not categorize into the different types of changes, the authors could present comparative metrics on all changes and not specific categories.

Most importantly, methods that reason in 3D such as (Taneja et al., 2011), [Adam et al., Objects can move: 3d change detection by geometric transformation consistency, ECCV 2022], [Palazzolo and Stachniss, Fast image-based geometric change detection given a
3d model, ICRA 2018] are not included in the experimental evaluation and are mostly not discussed in the related work. Such methods could be easily adopted to the given use case by using Mast3r to obtain the 3D models they reason on, which could result in strong baselines. Right now, the presented baselines do not integrate any kind of knowledge about the 3D scene.

[1] Kannan, Shyam Sundar, and Byung-Cheol Min. "Zeroscd: Zero-shot street scene change detection." 2025 IEEE International Conference on Robotics and Automation (ICRA). IEEE, 2025.

**Questions:**

Is the integration of Mast3R really necessary? Given that posed, calibrated images were used, using a simpler and more lightweight SfM method, wouldn’t the traditional render and compare lead to similar results? That would be an interesting experiment that would explore the trade-off between computational resource needed for the method and the success of the method.

Since the proposed method reasons in 3D, why are its results not also evaluated in 3D? Using an appropriate dataset, e.g., 3RScan [1] and extending the method by back-projecting the highlighted changes into 3D would also give interesting insights on the success of the method in the 3D environment.

[1] Wald, Johanna, et al. "Rio: 3d object instance re-localization in changing indoor environments." Proceedings of the IEEE/CVF International Conference on Computer Vision. 2019.

---

> ### Author Response · Authors · 2025-11-21
>
> Reviewer KMk1
>
> Thank you for the reviewer’s comments. We address each point in detail below.
> For your convenience added sections in the revision appear in blue.
>
> “The paper … identify objects … and track them using DEVA.”
>
> To clarify a possible misunderstanding, we do not use DEVA, all temporal propagation is performed using the video module of SAM2.
>
> “The novelty ... the methodologies applied in [1] are then deployed.”
>
> We respectfully note that this characterization is incorrect. Our work is not an extension of [1] and introduces several core innovations that go far beyond anything presented in that paper:
>
> We are the first to reframe image-based SCD as 3D stereo reconstruction across time, using 3D volumetric conflict to identify object-level changes in a zero-shot setting. Specifically, no prior method used 3D reconstruction to detect changes in an image pair.
> First method that can perform binary and multiclass change detection under zero-shot conditions on both image pairs and sets using the same pipeline, relying on the generability of our novel 3D conflict resolution approach, achieving state of the art results across all evaluated datasets.
> We are the first method to detect non-rigid changes from unaligned RGB image pairs under zero-shot conditions.
> Our method is 10x faster compared to related work when measured on multi-image scenes.
>
> [1] is limited to aligned image pairs, performs only binary detection, and cannot handle sets or unaligned inputs. It lacks all of the above capabilities. It is fundamentally different as it does not reason in 3D, but rely on feature map differencing.
>
> “More baselines ... including CYWS-3D ...”
>
> We chose benchmark datasets (Table 1) that are widely used in the SCD literature, enabling direct comparison with the maximum number of existing methods. These datasets include both indoor and outdoor real-world scenes exhibiting diverse visual changes (illumination, weather, seasonality, object motion, etc.).
>
> CYWS-3D, by contrast, is by part trained and evaluated on proprietary datasets. Nevertheless, following the reviewer’s suggestion, we conducted experiments on RC-3D, the same dataset used by CYWS-3D.
>
> As shown in Appendix A.2 and the newly added Table 9 in the new revision, our zero-shot method achieves 0.531 mAP using only RGB image pairs, surpassing CYWS-3D's performance with RGB input (0.41) and even RGB-D (0.50).

---

> > ### Comment · Reviewer_KMk1 · 2025-11-28
> >
> > Thank you for the detailed reply. I am sorry that it took me so long to get back to you.
> >
> > > To clarify a possible misunderstanding, we do not use DEVA, all temporal propagation is performed using the video module of SAM2.
> >
> > Thank you for the clarification. I was indeed wrong.
> >
> > > “The novelty ... the methodologies applied in [1] are then deployed.”
> >
> > The statement was that the proposed method relies on [1] for detecting changes in the synthesized views, which limits the novelty in the sense that using Mast3r to obtain poses and depth maps does not seem to be a novel contribution and that warping one image into another viewpoint via known depth values is also not a novel idea.
> >
> > > “More baselines ... including CYWS-3D ...”
> >
> > Thank you very much for adding this experiment. The experiment nicely demonstrates the value of the proposed approach compared to CYWS-3D.

---

> ### Author Response · Authors · 2025-11-21
>
> “Most importantly ... presented baselines do not integrate ... the 3D scene.”
>
> We respectfully note that this characterization is inaccurate. The cited methods assume complete 3D reconstructions as input, often requiring RGB-D scans or full 3D models at both timestamps. For example:
>
> Taneja et al.: requires a complete 3D model at t₀ and images at t₁.
>
> Palazzolo & Stachniss: requires a prior 3D model and ~5 registered views per scene.
>
> Adam et al.: requires two full 3D models, at t₀ and t₁.
>
> Our method, in contrast, operates on RGB-only image pairs or sets, with no 3D model inputs, no depth, and no scan alignment. Their problem setting is fundamentally not comparable to ours. To highlight said difference, the core ability of our method, change detection given an image-pair input, is an impossible task for the mentioned methods
>
> Regarding the reviewer’s suggestion to use MASt3R the 3D models these methods rely on, we respectfully hold the opinion that comparing to papers that skipped the 3d reconstruction step would be biased in their favor. Instead, we compare against NeRF-CD, 3DGS-CD, and Gaussian Difference which are 3D-centric methods that use only RGB and reconstruct geometry as part of their pipeline. We clearly outperform all three (see Table 4 and Appendix A.2), under zero-shot conditions, while requiring no auxiliary images, and being 10–100× faster as seen in the newly added Table 5 available in the revision.
>
> We have added further clarification in the related work section regarding this class of related methods.
>
> “Is the integration ...”
>
> As stated in Table 1, the majority of datasets we evaluate on contain unaligned, uncalibrated images.
> Specifically, in our experiments we evaluated on 4 datasets, out of which 3 are characterized by such unaligned, non-calibrated images with unknown camera positions. As such, MAST3R is required in order to reproject the images into the other view camera, as demonstrated by the ablations we performed, as seen in Table 10. For the VL-CMU-CD dataset which contains aligned images, we skipped the 3D reconstruction step as it was not needed and MAST3R was not used, lessening the computational resource requirements. Having said that, As seen in the newly added Table 5 available in the new revision, we are 10-100x faster than related work when measured on multi-image scenes.
>
> “Since the proposed method ... Using an appropriate dataset, e.g., 3RScan ..”
>
> We respectfully disagree with the applicability of this suggestion. Our method operates purely on RGB image input and is designed for 2D semantic change detection, not for object pose estimation or 3D instance re-localization. Our method does not consume or predict 3D object poses; it does not assume access to registered 3D scans at t₀ and t₁. The 3RScan dataset is constructed for an entirely different task: it provides RGB-D scans and object transformation annotations, which are incompatible with both our input modality and output format.

---

> > ### Comment · Reviewer_KMk1 · 2025-11-28
> > **Applicability of Taneja et al. as a baseline**
> >
> > Regarding the applicability of the three baselines to the setting of the paper under submission:
> >
> > > Taneja et al.: requires a complete 3D model at t₀ and images at t₁.
> >
> > Given two images taken at time t1 and a 3D model constructed before t1, Taneja et al. are interested in determining which parts of the 3D model are outdated / have changed. Given poses for the two images, Taneja et al. warp the first image into the second one using depth maps obtained by rendering the 3D model from the known poses. They then photometrically compare the rendered to the actual image. Since both images are taken at the same point in time, photometric differences between the two images are then caused by changes in the underlying scene geometry, as outdated geometry will cause pixels to be projected into the wrong location in the second image. Obviously, there will be spurious detections due to inaccuracies in the camera poses or inaccuracies in the 3D model, as well as occluded regions. In addition, some changes will be missed as they are not directly visible (e.g., the back of the statue in Fig. 1 of Taneja et al.). Thus, after discarding some potential changes based on semantic segmentations, Taneja et al. solve a graph cut problem in 3D to determine changed regions (where the graph cut formulation allows them to both propagate initial change detections (based on photometric differences) to parts of the scene where no change was directly observed but that are close in 3D, and to handle spurious change detections robustly).
> >
> > Looking at the figures provided by Taneja et al. (both in the original ICCV 2011 paper and the journal extended published in TPAMI), the approach from Taneja et al. can be applied on incomplete 3D models (where not all parts of the scene are reconstructed due to missing observations or where scene parts are not reconstructed (especially thin structures)). In the journal version, they also consider a scenario where the 3D model is obtained from cadastral maps, i.e., was not reconstructed from images, and only contains rather coarse geometry. As such, the claim that "The cited methods assume complete 3D reconstructions as input" is not correct for the approach by Taneja et al. (intuitively, only the parts visible in the two images need to be reconstructed sufficiently well to allow image warping (the same assumption is made in the paper under review), parts not observed in the images will (in most cases) not have an impact).
> >
> > The approach from Taneja et al. can be directly applied to a scenario where a 3D model and an image are captured at time t0 and another image is captured at time t1. Warping the first image into the second image using the 3D model (actually, a depth map for the first image would be sufficient), changes will be visible through photometric differences. Semantic filtering and graph cut in 3D can then be applied to propagate initial changes and handle spurious changes. Compared to the setting considered in Taneja et al., the photometric consistency assumption does not necessarily hold as the images are taken at different points in time. Still, the graph cut might be robust enough to handle this issue. Alternatively, using segmentation consistency instead of photometric consistency (as done in [Adam et al., Has anything changed? 3d change detection by 2d segmentation masks, arXiv:2312.01148]) or feature-metric differences could readily solve the issue.
> >
> > Given two images, their relative poses, and depth maps, I do not see why the approach from Taneja et al. cannot be readily (without any change) be employed to the problem considered in the paper under review. Given that the proposed approach uses Mast3r out-of-the-box to obtain this information, I don't see why it couldn't be provided to the method by Taneja et al. Given that the proposed approach was designed and tuned to handle the characteristics of the depth maps provided by Mast3r, while the approach by Taneja et al. wasn't, I don't see how the argument that "comparing to papers that skipped the 3d reconstruction step would be biased in their favor" would apply in this case. If anything, I would expect a bias in favor of the proposed approach.

---

> > > ### Comment · Reviewer_KMk1 · 2025-11-28
> > > **Applicability of Palazzolo & Stachniss as a baseline**
> > >
> > > > Palazzolo & Stachniss: requires a prior 3D model and ~5 registered views per scene.
> > >
> > > As Taneja et al., Palazzolo & Stachniss (PS) use a 3D model to warp one image into another via a depth map obtained by rendering the 3D model. Again, photometric consistency is used to detect potential changes in the geometry of the 3D model between the time the model was build and the new images were captured. In the case where the geometry has changed, this projection can lead to duplicates of the changed region as the region containing the change in the second image will not be consistent with the corresponding region in the rendered image. At the same time, the change in the first image will be propagated to another region in the second image to using incorrect depths (see Fig. 2 in PS for an illustration). Taneja et al. handle such inconsistencies through a graph-cut-based optimization problem. PS instead use multiple images to resolve this ambiguity, which further allows them to obtain an estimate of the changed region in 3D space via triangulation.
> > >
> > > Obviously, the parts about projecting one image to another and measuring photometric differences can also be applied in a two-image scenario, where both images are captured at different times and depth maps and relative poses are available, e.g., from applying Mast3r on the two images. Since depth maps are available, I don't think that additional images are needed to resolve the ambiguity in the photometric differences. Rather, the changed 3D parts between the two images can be directly detected by taking looking up the depth values in regions with large photometric differences.
> > >
> > >
> > > Again, the 3D model only needs to be reasonably complete in the viewpoints covered by the images. I.e., using relative poses and depth maps from Mast3r would allow the approach by PS to be applied in the two-view setting considered in this work. Again, the photometric consistency assumption might be violated, but the approach can still be applied and used as a baseline. In addition, the segmentation and data association step from PS might be sufficient to handle the arising issues (the feature correspondences readily provided by Mast3r can help with data association).
> > >
> > > In short, I think a simplified version of PS (which drops the stage where multiple images are used for disambigution) can be directly used in the problem setting considered in this work and should thus be included as a baseline.

---

### Author Response · Authors · 2025-12-03
**Discussion summary**

Dear AC,

We appreciate your time and effort, and summarize the rebuttal discussion for your convenience. We start with the current status and follow with the main concerns that were raised during the rebuttal, along with our responses to them.

Status:

We would like to comment that our method can handle both pairwise and multi-image settings with binary and multiclass labels, and was tested on five diverse datasets spanning real-world and synthetic, indoor and outdoor scenes. We consistently outperform all baselines, including supervised ones, across all datasets except one with known annotation errors, and our performance is between 10 to 100 times faster. To our knowledge, ours is the only method capable of handling such a wide range of datasets and input configurations, demonstrating exceptional robustness and flexibility.

Initially, two reviewers were positive about the paper and gave us a score of 6, 14W9 and rCUL, and two that gave us a score of 4, x35Y and KMk1, who asked questions about additional baselines, runtime comparisons and 3D model performance which we addressed with a series of experiments, ablation studies and explanations. Please find below a summary with regards to each reviewer's discussion.

Reviewer 14W9 gave us a score of 6, and we have addressed the concerns raised by the reviewer and thus were hoping the reviewer would raise their score. . We hope that the AC can go over the rebuttals and validate that we indeed resolved the concerns of 14W9, and thus accept their assessment.

Reviewer rCUL gave a score of 6 and has not yet responded. The reviewer’s  concerns about model choice, 3D reconstruction novelty, and parameter settings were addressed by clarifying our modular design, novel use of reconstruction as a diagnostic tool, and an additional ablation on detection thresholds. We hope these responses resolve their feedback.

Reviewer x35Y gave us a score of 4, and was concerned about runtime, applicability, 3D model performance, and our depth-aware conflict resolution. We provided runtime analysis demonstrating we are 10-100 times faster than baselines, clarified SCD’s offline nature and general applicability, conducted a 3D model ablation, and included a new figure illustrating conflict resolution. While the reviewer hasn't answered prior to the discussion freeze, we hope we managed to address all of their concerns as well.

Reviewer KMk1 gave a score of 4 before the discussion ended. We addressed several concerns with new experiments, outperforming their suggested baseline, and demonstrating our method’s ability to provide predictions in 3D format. The reviewer suggested comparisons to other baselines which require 3D scenes as input. We believe that such a scenario - having a 3D scene as input is a fundamentally different problem setting, and would require major reengineering to adapt, which is beyond the scope of our paper that deals with image input. Regarding novelty compared to another baseline, we explained our novel contribution lies in reframing SCD as a reconstruction problem from time-separated views, as opposed to the feature matching approach proposed by the baseline.

During the rebuttal, we uploaded a revised version of our paper with changes and additions requested by the reviewers. Changes are marked in blue for your convenience. The main changes and additions to the paper were:

- New experimental results over the RC-3D dataset (Table 9 and Appendix A.2).
- New experimental results on a scene from the 3RScan dataset (Figure 7 and Appendix A.2)
- New Run time analysis and comparisons to other baselines (Table 5).
- New Ablation study, exploring the relationship between 3D reconstruction performance and generality, and multiclass change detection performance (Figure 10).
- New Ablation study regarding the cross-view voting component on the 3DGS-CD dataset (Table 11).
- New analysis exploring the detection thresholds of SAM2 and their effect on performance on the VL-CMU-CD dataset (Table 13).
- Additional illustration of the depth-filtering process with renderings and qualitative visualizations to aid the reader’s understanding of this stage of our pipeline (Figure 8).
- Details regarding motivation behind design and foundational model choices.

---

> ### Author Response · Authors · 2025-12-03
> **More details on the discussion thus far and our responses:**
>
> More details on the discussion thus far and our responses:
>
> Runtime and resources – Reviewers x35Y and rCUL asked about runtime and computational complexity. We added Table 5 with runtime comparisons, showing our method is 10–100× faster on multi-image scenes.
>
> Model choice – Reviewer rCUL asked about our choice of foundational models. We clarified that our pipeline is modular and model-agnostic, and other models can be used. Our selections were based on state-of-the-art performance.
>
> Effect of the 3D reconstruction model – Reviewers rCUL and 14W9 asked about its role and novelty. We clarified that 4 of 5 datasets contain unaligned, uncalibrated images, making MAST3R essential for reprojection due to the nature of the ground truth. Figure 10 presents an ablation on reconstruction quality vs. change detection. While prior work uses 3D reconstruction to rebuild scenes, we exploit the reconstruction process itself to reveal structural inconsistencies between time-separated images—allowing effective detection even from a single image pair.
>
> Thresholds and parameters – Reviewer 14W9 asked about the SSIM threshold and SAM2 parameters. We clarified that a single data-driven SSIM threshold is used across all examples, while SAM2 parameters remain nearly identical across datasets (Appendix A.5, Table 12). Table 13 was added to present an analysis of SAM2 detection threshold effects.
>
> Additional datasets and baselines – Reviewer KMk1 requested comparisons to more methods. We added RC-3D results (Table 9), where our zero-shot RGB approach outperforms the supervised RGB-D baseline CYWS-3D. We clarified that the suggested 3D methods require 3D inputs and outputs, and solve a fundamentally different problem than ours. Regarding 3RScan, we explained that our method targets 2D semantic change detection, while 3RScan provides 3D instance re-localization annotations as ground truth, a modality we can not evaluate on. Despite that, we added Figure 7, demonstrating our ability on a 3RScan scene to provide predictions in 3D format.
> Novelty – Reviewer KMk1 compared our work to ZeroSCD: Zero-shot Street Scene Change Detection. We clarified that our approach is fundamentally different: we do not rely on VPR-based pretraining or feature matching. While we both methods use SAM in the pipeline, it is  done for different reasons: mask boundary refinement in the baseline compared to object detection and temporal mask propagation in ours. Our novelty lies in leveraging the 3D reconstruction process itself as a diagnostic tool - using reconstruction conflicts from time-separated views to reveal changes, even in unaligned scenarios.
>
> We highly appreciate your consideration. Thank you.

---

### Meta-Review · Area_Chair_QEMn · 2026-01-09

**Summary:**

This paper proposes GOLDILOCS as a zero-shot, pose-agnostic object-level semantic change detection pipeline that leverages 3D reconstruction conflicts, novel-view rendering, and class-agnostic segmentation/tracking (SAM2), covering both image-pair and multi-view settings and (where available) multiclass labels. Reviewers generally agree that the problem is important and the method is well-motivated with good empirical results across multiple datasets. The main concerns were (i) novelty relative to prior render/warp-and-compare pipelines and the role/necessity of MASt3R, (ii) baseline coverage (especially classical 3D-model-based change detection), and (iii) efficiency/implementation details as well as the parameter sensitivity.

Overall, reviewers found the method technically sound and empirically strong, with the rebuttal addressing key concerns and pushing it above the acceptance threshold.

**Reviewer Concerns:**

- Addressed by rebuttal/revision:

-Runtime / resource concerns (x35Y, rCUL): authors added runtime comparisons and memory/time figures, arguing favorable efficiency vs. relevant baselines and clarifying offline SCD use-cases.

-Ablations / clarity (14W9, x35Y): added analyses on reconstruction quality vs. SCD performance, parameter/threshold sensitivity.

-Baseline request for CYWS-3D (KMk1): authors added RC-3D experiments and the reviewer acknowledged this addition as valuable.

- Still Partially addressed:

-Novelty framing (KMk1): while the pipeline is effective, the core warping/rendering comparison enabled by depth/poses is not entirely new; novelty rests mainly in the “reconstruction-as-diagnostic/conflict” formulation and broad applicability.

-Missing classical 3D-model-based baselines (KMk1): the applicability discussion is substantive (and the reviewer argues these methods can be adapted using MASt3R outputs). Even if direct comparability is debatable, the absence remains a limitation.

-Broader evaluation on more challenging real-world dynamics (14W9): acknowledged as future work/limitations.

**Reviewer Scores:**

- Reviewer 14W9: 6 (confidence 3). Posted a positive post-rebuttal assessment stating concerns were satisfactorily addressed, but did not post a score change.
- Reviewer rCUL: 6 (confidence 3). No recorded discussion update / score change after rebuttal in the provided text.
- Reviewer x35Y: 4 (confidence 4). No recorded score change after rebuttal in the provided text.
- Reviewer KMk1: 4 (confidence 4). Engaged in discussion; acknowledged added CYWS-3D experiment and clarified misunderstandings, but raised substantive follow-up about applicability of classic baselines; no recorded score change.
- Missing review(s): Not available at decision time.

---

### Decision · Program_Chairs · 2026-01-26

Accept (Poster)